# Adaptation and compensation in a bacterial gene regulatory network evolving under antibiotic selection

Vishwa Patel[1,2], Nishad Matange[3]*

[1]Dr. Vikram Sarabhai Institute of Cell and Molecular Biology, The Maharaja Sayajirao University of Baroda, Vadodara, India; [2]Department of Microbiology and Cell Biology, Indian Institute of Science, Bangalore, India; [3]Indian Institute of Science Education and Research (IISER), Pune, India

**Abstract** Gene regulatory networks allow organisms to generate coordinated responses to environmental challenges. In bacteria, regulatory networks are re-wired and re-purposed during evolution, though the relationship between selection pressures and evolutionary change is poorly understood. In this study, we discover that the early evolutionary response of *Escherichia coli* to the antibiotic trimethoprim involves derepression of PhoPQ signaling, an $Mg^{2+}$-sensitive two-component system, by inactivation of the MgrB feedback-regulatory protein. We report that derepression of PhoPQ confers trimethoprim-tolerance to *E. coli* by hitherto unrecognized transcriptional upregulation of dihydrofolate reductase (DHFR), target of trimethoprim. As a result, mutations in *mgrB* precede and facilitate the evolution of drug resistance. Using laboratory evolution, genome sequencing, and mutation re-construction, we show that populations of *E. coli* challenged with trimethoprim are faced with the evolutionary 'choice' of transitioning from tolerant to resistant by mutations in DHFR, or compensating for the fitness costs of PhoPQ derepression by inactivating the RpoS sigma factor, itself a PhoPQ-target. Outcomes at this evolutionary branch-point are determined by the strength of antibiotic selection, such that high pressures favor resistance, while low pressures favor cost compensation. Our results relate evolutionary changes in bacterial gene regulatory networks to strength of selection and provide mechanistic evidence to substantiate this link.

*For correspondence: nishad@iiserpune.ac.in

## Introduction

The relationship between genotype and phenotype is complex and much of modern genetics aims at understanding how it is established and maintained. In cellular organisms, this relationship is dictated in large part by the regulation of gene expression in response to the internal and external environment (*Hill et al., 2021*; *Kemble et al., 2019*; *Kim and Przytycka, 2012*). By regulating gene expression, organisms select which traits are expressed, enabling a wide palette of phenotypes to be generated from the same genotype. Not surprisingly, the functioning and evolution of mechanisms that regulate gene expression are central questions at the cusp of genetics and evolutionary biology with implications for diverse phenomena such as development (*Levine and Davidson, 2005*), differentiation (*Levine and Davidson, 2005*), and disease (*Singh et al., 2018*).

In bacteria, like in more complex organisms, gene regulatory proteins, such as transcription factors and signal transducers are organized into networks (*Shen-Orr et al., 2002*). The predominant sensory modules in bacteria are two-component signaling systems, which are made up of membrane-bound receptors and cognate cytosolic response regulators (*Perraud et al., 1999*). Upon activation of the receptor by an appropriate ligand, a phospho-relay between the two components is initiated, which changes the phosphorylation state of the response regulator. Response regulators are often

transcription factors themselves, which bind to DNA upon phosphorylation and produce changes in gene expression (*Perraud et al., 1999*). Individual two-component pathways are integrated into networks by 'connector' proteins, and also by cross-activation (*Mitrophanov and Groisman, 2008*). Further, two-component systems are integrated with other signaling molecules in bacteria such as regulatory RNAs (*Valverde and Haas, 2008*) and sigma factors (*Hengge, 2008*).

Gene regulatory networks orchestrate a number of physiological functions in bacteria such as sporulation (*Stephenson and Lewis, 2005*), virulence (*Beier and Gross, 2006*), dormancy (*Sivaramakrishnan and de Montellano, 2013*), and tolerance to stresses like acid (*Xu et al., 2020*), heavy metal (*Perron et al., 2004*), and osmolarity (*Yuan et al., 2017*). More recently, the role of bacterial gene regulatory networks in antimicrobial resistance is beginning to emerge (*Tierney and Rather, 2019*). VanRS (vancomycin resistance, *Staphylococcus aureus*) (*Jansen et al., 2007*), BaeRS (ciprofloxacin resistance, *Salmonella typhimurium*) (*Guerrero et al., 2013*), PhoPQ (*Bader et al., 2005*; *Yadavalli et al., 2016*), PmrAB (*Gunn, 2008*) (antimicrobial peptide resistance, *Salmonella* and *Escherichia coli*), MtrAB (multidrug resistance, *Mycobacterium tuberculosis*) (*Nguyen et al., 2010*), and CroRS (β-lactam resistance, *Enterococcus faecalis*) (*Comenge et al., 2003*) are some of the two-component systems that mediate drug resistance in bacteria. Two-component systems modulate antibiotic resistance by regulating the expression of downstream effectors, such as efflux pumps, porins, or enzymes that modify the cell envelope (*Tierney and Rather, 2019*). Not surprisingly, two-component sensor kinases have also been proposed as potential targets for the design of novel anti-microbials (*Bem et al., 2015*).

The repertoire of two-component systems that bacteria possess depends on environment and life history. For instance, *Myxococcus xanthus* that faces rapidly changing, diverse environments codes for more than 250 different two-component systems (*Shi et al., 2008*). Further, rewiring of two-component systems during evolution is also seen among bacteria. For example, the regulons of $Mg^{2+}$-sensitive PhoPQ systems from closely related members of *Enterobacteriaceae* show several species-specific features. Most notably, *Salmonella* and *Yersinia* have only a marginal overlap in PhoP-regulons, even though they share a common ancestor (*Perez et al., 2009*). Similarly, co-option of existing regulatory modules for the expression of genes coded by horizontally acquired pathogenicity islands in *Salmonella* has also been described (*Palmer et al., 2019*). Thus, two-component systems and their associated transcriptional networks are likely to be ready substrates for evolution to suit the unique demands of different growth environments.

Current understanding of the evolution of two-component signaling systems and their associated gene regulatory networks comes from two main approaches. First of these relies on the comparison of orthologous genes from related bacterial species (*Perez et al., 2009*; *Monsieurs et al., 2005*). The second uses genetically engineered bacterial strains to understand general principles governing the specificity of receptor-regulator pairs and the potential for cross-activation in two-component systems (*Schmidl et al., 2019*). While invaluable insight has emerged from both approaches, the direct link between evolutionary changes in gene regulatory networks and selection pressures remains elusive. In this study, we use laboratory evolution to identify a specific environmental factor, that is, the antibiotic trimethoprim, as a selective pressure that leads to derepression of the PhoPQ signaling pathway in *E. coli*. This adaptation is mediated by loss-of-function mutations at the *mgrB* locus, which codes for a feedback attenuator of PhoPQ signaling. Intrinsic resistance to trimethoprim in *E. coli* is thought to be acquired primarily by mutations in the drug-target dihydrofolate reductase (DHFR), coded by the *folA* gene (*Toprak et al., 2011*; *Watson et al., 2007*). We find that mutations in DHFR are typically preceded by mutations at the *mgrB* locus and together, they have a synergistic effect on the level of trimethoprim resistance. We use this system to ask how the costs of adaptation by perturbation of gene regulatory networks are compensated over long-term evolution and identify the key role that strength of selection plays in determining the adaptive trajectory that bacteria take upon antibiotic challenge.

## Results

### Mutation at the *mgrB* locus rather than *folA* is an early evolutionary response of *E. coli* challenged with trimethoprim

We had earlier evolved trimethoprim-resistance in *E. coli* K-12 MG1655 by serially passaging bacteria in drug-supplemented medium for ~25 generations (*Matange et al., 2018*). From these experiments,

10 independently evolved trimethoprim-resistant isolates (designated TMPR1-10), with drug IC50 ranging from 6- to 300-fold over wild-type were obtained (*Figure 1A*). Unexpectedly, only 3 of the 10 trimethoprim-resistant isolates (TMPR 1, 3, and 5) had mutations in the coding region of the *folA* gene, which suggested that early adaptation to trimethoprim did not necessarily involve mutations in the drug target itself (*Figure 1A*). We, therefore, sequenced the genomes of 5 of the 10 isolates (TMPR1–5) to identify other genes that may contribute to trimethoprim resistance. Only one locus, coding for the *mgrB* feedback regulator protein, consistently harbored mutations in all five isolates (*Table 1*). Four of the five isolates (TMPR1–4) harbored IS-element insertions in the promoter of *mgrB* between the PhoP-binding site and the translational start site of the gene (*Figure 1A*, *Table 1*). Similar insertions of IS-elements in the promoter of *mgrB* attenuate its expression levels in other bacteria such as *Klebsiella* (*Pitt et al., 2018*; *Uz Zaman et al., 2018*). The fifth isolate (TMPR5) harbored a single nucleotide deletion in the coding region of *mgrB* that resulted in a frame-shift and altered sequence of the C-terminal domain of the protein (*Figure 1A*, *Table 1*). The C-terminus of MgrB is necessary for its inhibitory activity against PhoQ and mutations in this region alter the ability of MgrB to bind to PhoQ (*Yadavalli et al., 2020*). Sanger sequencing of the *mgrB* gene and its promoter from isolates TMPR6–10 confirmed the presence of similar mutations in these bacteria as well (*Figure 1A*), suggesting that lower MgrB expression or activity may be advantageous for *E. coli* in the presence of trimethoprim.

We tested this hypothesis using an independently generated *ΔmgrB* strain and found that loss of MgrB enhanced trimethoprim IC50 by ~3 -fold (*Figure 1B and C*). Thus, mutation-driven loss of *mgrB*, rather than mutations in the *folA* gene, was the predominant early adaptive change in *E. coli* challenged with trimethoprim.

## Loss of *mgrB* confers trimethoprim tolerance by PhoP-dependent upregulation of DHFR expression

To better understand how loss of *mgrB* reduced the effectiveness of trimethoprim against *E. coli*, we further characterized the *mgrB* knock-out strain. *E. coli ΔmgrB* had lower relative fitness than wild-type in antibiotic-free media ($w_{ΔmgrB}$=0.86±0.12). However, sub-inhibitory concentration (300 ng/ml) trimethoprim in growth media enhanced relative fitness of the mutant ($w_{ΔmgrB}$=1.4±0.08). Concomitantly, *E. coli ΔmgrB* was selected over wild-type in co-cultures when trimethoprim was added to growth media. The competitive advantage of *E. coli ΔmgrB* was apparent even when mixing ratios were biased in favor of wild-type (*Figure 1D*).

Interestingly, there was no detectable difference in trimethoprim minimum inhibitory concentration (MIC) between wild-type and mutant in broth dilution assay or E-test (*Figure 1B and E*). However, *mgrB*-deficient *E. coli* had higher colony-forming efficiency than wild-type on trimethoprim-supplemented solid media even at concentrations close to MIC (*Figure 1E*). Similarly, relatively fewer cells of *E. coli ΔmgrB* were required to colonize liquid growth media containing inhibitory concentrations of trimethoprim (*Figure 1—figure supplement 1*). Thus, loss of *mgrB* allowed better survival of *E. coli* over a wide range of trimethoprim concentrations, without a detectable increase in MIC. Therefore, we concluded that *mgrB*-deficiency conferred trimethoprim tolerance, rather than resistance, to *E. coli*. Tolerance has been understood recently as a change in the death rate of bacteria by bactericidal antibiotics without change in MIC. However, since trimethoprim is a bacteriostatic antibiotic, we define 'tolerance' for trimethoprim as reduced susceptibility to the antibiotic, without substantial change in MIC.

The primary role of the MgrB protein in *E. coli* is to attenuate PhoQP signaling through negative feedback (*Lippa and Goulian, 2009*; *Salazar et al., 2016*). Thus, PhoQP signaling is derepressed in an *mgrB* knockout strain (*Lippa and Goulian, 2009*; *Salazar et al., 2016*). In line with this role, deletion of *phoP* re-sensitized *E. coli ΔmgrB* to trimethoprim, while in a wild-type background had no detectable effect (*Figure 1C, D and E*, *Figure 1—figure supplement 1*). PhoP directly activates transcription of close to 50 genes in *E. coli* with diverse functions (*Minagawa et al., 2003*). Among these is *iraM*, which protects the stress-responsive sigma factor RpoS from degradation (*Xu et al., 2019*). Arguing that trimethoprim tolerance in the *mgrB*-knockout could be a result of enhanced general stress-response pathway, we generated knockout strains of *iraM* and *rpoS* in a *ΔmgrB* background. However, neither deletion altered trimethoprim susceptibility of *ΔmgrB*, ruling out this possibility (*Figure 1C, D and E*, *Figure 1—figure supplement 1*).

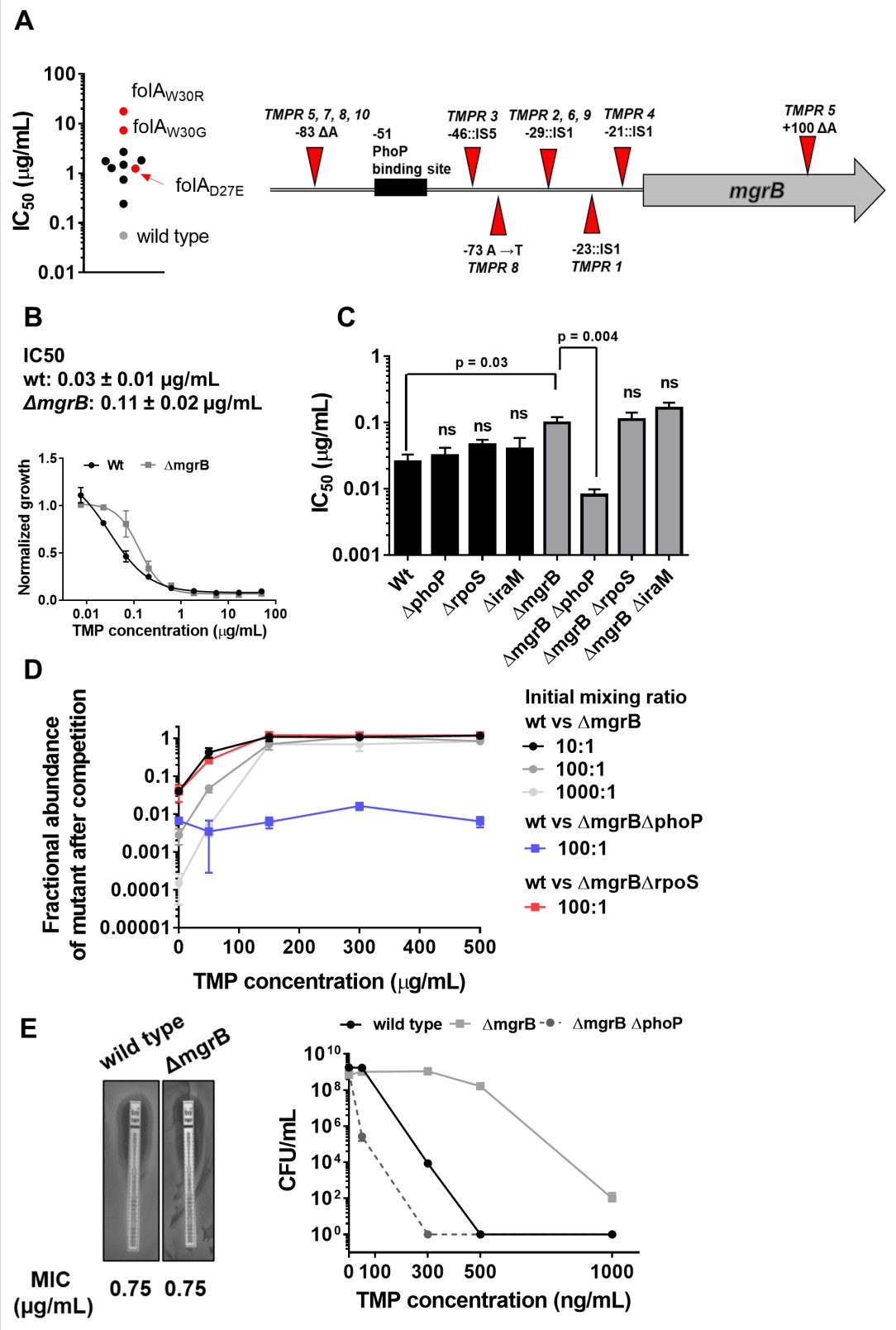

Figure 1. Loss of *mgrB* confers trimethoprim tolerance to *Escherichia coli*. (**A**) Left panel. Trimethoprim IC$_{50}$ values of resistant isolates derived from *E. coli* K-12 MG1655 (wild-type, gray) by laboratory evolution. Mean IC$_{50}$ values from three independent measurements are plotted. Isolates without mutations in the *folA* locus are represented by black circles. Isolates with mutations in *folA* are represented as red circles, and the identified mutation is indicated. Right panel. Diagrammatic representation of the *mgrB* gene and its promoter showing mutations identified in trimethoprim resistant

*Figure 1 continued on next page*

*Figure 1 continued*

isolates TMPR1–5 by genome sequencing or Sanger sequencing from TMPR6–10. The PhoP-binding site in the *mgrB* promoter is shown as a black box. Location of each mutation is calculated as base pairs from the translation start site. (**B**) Growth of wild-type (black) and Δ*mgrB* (gray) *E. coli* in varying concentrations of trimethoprim, normalized to growth in drug-free medium. Each data point represents mean± SD from three independent experiments. $IC_{50}$ values represent mean± SEM obtained after curve fitting. (**C**) $IC_{50}$ values of trimethoprim for wild-type or mutant *E. coli*. Each bar represents mean± SEM from three independent measurements. Statistical significance was tested using a Student's t-test. A p-value <0.05 was considered significant (shown), while p-values≥0.05 were considered non-significant (ns). (**D**) Competition between *E. coli* wild-type and Δ*mgrB* in increasing concentrations of trimethoprim starting at the indicated mixing ratios. The fraction of mutant bacteria (CFUs of mutant/total CFUs) in each mixed culture after ~9 generations of competition are plotted (mean± SD from three independent experiments). Results of competition between wild-type and Δ*mgrB*Δ*rpoS* or Δ*mgrB*Δ*phoP* strains at an initial mixing ratio of 100:1 are also shown. (**E**) Left panel. MIC values of trimethoprim for wild-type and *E. coli* Δ*mgrB* calculated from E-tests. Right panel. Colony formation of wild-type and Δ*mgrB* on solid media supplemented with indicated concentrations of trimethoprim. Each point represents mean± SD from three independent experiments.

The online version of this article includes the following figure supplement(s) for figure 1:

**Figure supplement 1.** Colonization of indicated *Escherichia coli* strains in growth media supplemented with increasing trimethoprim concentrations ( 5, 1, 0.5 and 0 µg/mL) starting at the indicated cell densities (X-axis). Optical density of cultures was measured after 24 hr of growth. Mean± SD is plotted.

To identify the target of PhoP that led to trimethoprim-tolerance, we turned to the transcriptomics data published by *Xu et al., 2019* that compared gene expression profiles of wild-type and *mgrB*-deficient *E. coli*. Among over 500 genes upregulated in *mgrB*-deficient *E. coli*, 97 are known targets of RpoS and hence could be excluded (*Figure 2—figure supplement 1*). Curiously, *folA* transcript levels were reported to be enhanced by ~3 -fold in an *mgrB*-deficient strain (*Xu et al., 2019*). The *folA* gene codes for DHFR, target of trimethoprim, and its overexpression results in trimethoprim resistance (*Flensburg and Sköld, 1987*). We found that DHFR protein levels were indeed elevated in an *mgrB*

**Table 1.** Summary of mutations in trimethoprim resistant isolates TMPR1–5 identified by genome sequencing.

| Locus | Description | TMPR1 | TMPR2 | TMPR3 | TMPR4 | TMPR5 |
|---|---|---|---|---|---|---|
| *Single nucleotide changes* | | | | | | |
| *folA* | Dihydrofolate reductase | Asp27Glu | – | Trp30Gly | – | Trp30Arg |
| *ycaQ* | Inter-strand DNA crosslink repair glycosylase | – | – | Trp71* | – | – |
| *yagF* | D-xylonate dehydratase | – | – | – | – | Ser557Pro |
| *yagH* | Putative xylosidase/arabinosidase | – | – | – | – | Val397Met |
| *mgrB* | PhoQ kinase inhibitor | – | – | – | – | Deletion + 100::ΔA |
| *hexR* | DNA-binding transcriptional repressor | – | – | – | – | Ala143Val |
| *deoA* | Thymidine phosphorylase | Insertion + 359::A | – | Insertion + 375::A | - | - |
| *dnaA* | Chromosomal replication initiator protein | – | – | – | Asp280Gly | – |
| *Insertion elements and large rearrangements* | | | | | | |
| *mcrC* | 5-methylcytosine-specific restriction enzyme subunit | – | – | + 969::IS1A | – | – |
| *ybcM* | Putative DNA-binding transcriptional regulator | + 493::IS1A | – | – | – | – |
| *mgrB* | PhoQ kinase inhibitor | –23::IS1A (772 bp insertion) | –29::IS1A (772 bp insertion) | –46::IS5 ( ~ 1100 bp insertion) | –21::IS1A (772 bp insertion) | – |
| Partial genome duplication | – | – | – | – | 2× (~1–7,00,000) | – |

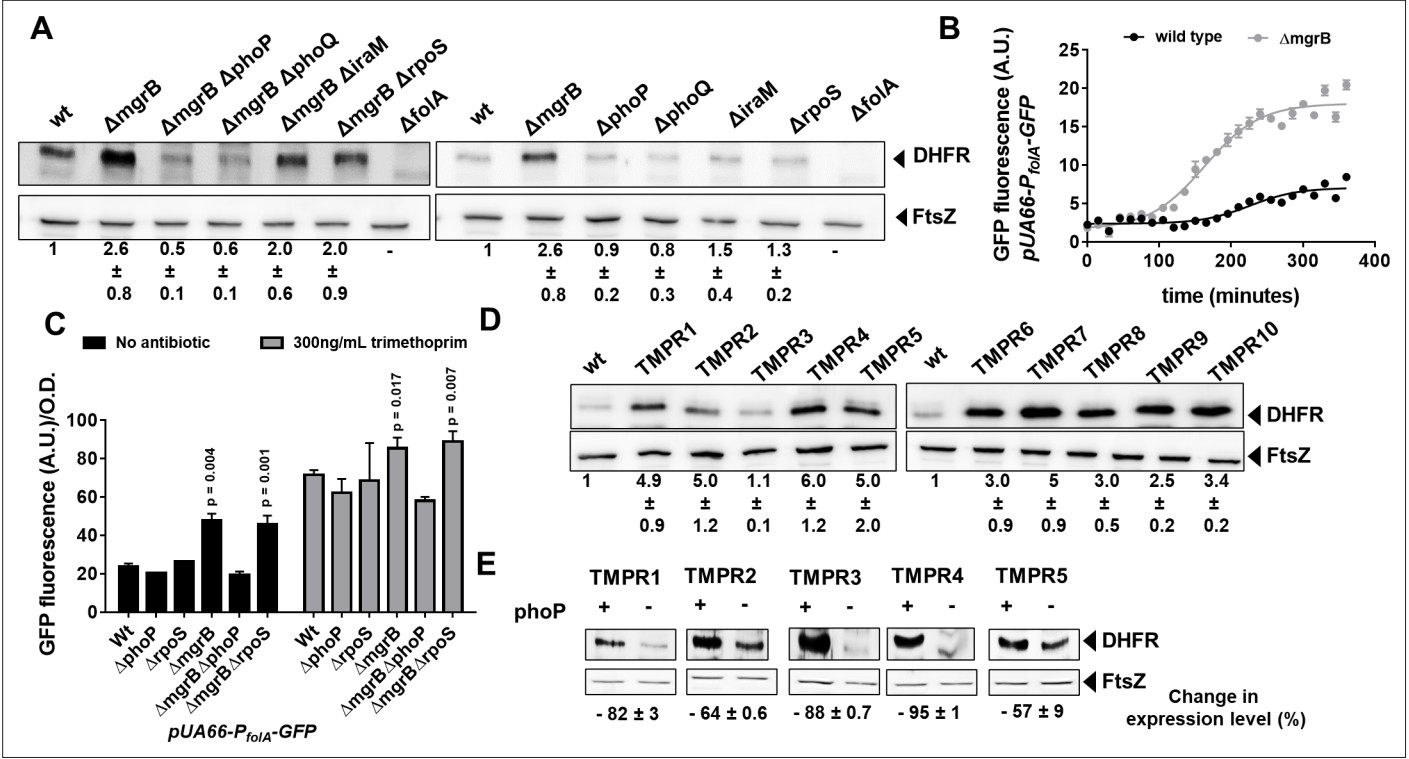

**Figure 2.** Enhanced expression of DHFR in *mgrB*-deficient *Escherichia coli*. (**A**) Lysates (5 µg total protein) of wild-type or mutant *E. coli* were subjected to immunoblotting using anti-DHFR polyclonal antibody. FtsZ was used as a loading control. Data shown are representative of three independent replicates. Fold change in expression level of DHFR relative to wild-type (normalized to 1) was calculated by quantitating band intensities and normalizing to FtsZ. Values shown are mean± SD from three independent experiments. (**B**) Activity of the *folA* promoter ($P_{folA}$) in *E. coli* wild-type or *ΔmgrB* monitored over growth using a GFP reporter gene (arbitrary units, A.U.). Each point represents mean± SD from three replicates. (**C**) Peak fluorescence normalized to optical density for indicated strains harboring the $P_{fol}$-GFP reporter plasmid. Mean± SD from three independent experiments is plotted. Statistical significance was tested using the Student's t-test and a p-value of <0.05 was considered significant (p-value indicated). Promoter activity was measured in drug-free medium or in the presence of 300 ng/ml trimethoprim. (**D**) Lysates (5 µg total protein) of *E. coli* wild-type or trimethoprim resistant isolates (TMPR1–10) were subjected to immunoblotting using anti-DHFR polyclonal antibody. FtsZ was used as a loading control. Data shown are representative of three independent replicates. Fold change in expression level of DHFR relative to wild-type (normalized to 1) was calculated by quantitating band intensities and normalizing to FtsZ. Values shown are mean± SD from three independent experiments. (**E**) Lysates (5 µg total protein) of trimethoprim-resistant isolates (TMPR1–5) and their *ΔphoP* derivatives were subjected to immunoblotting using anti-DHFR polyclonal antibody. FtsZ was used as a loading control. Data shown are representative of three independent replicates. Reduction in expression level of DHFR upon deletion of phoP was calculated by quantitating band intensities and normalizing to FtsZ. Percent reduction value shown is mean± SD from three independent experiments. DHFR, dihydrofolate reductase.

The online version of this article includes the following figure supplement(s) for figure 2:

**Figure supplement 1.** Diagrammatic representation of upregulated genes in *Escherichia coli ΔmgrB*.

**Figure supplement 2.** Electrophoretic mobility shift assay (EMSA) to test in vitro binding of phosphorylated PhoP (PhoP-P) to the promoter of *folA* ($P_{folA}$).

**Figure supplement 3.** DHFR protein levels measured in wild-type and mutant strains (indicated) by immunoblotting with anti-DHFR antibody.

knockout strain (*Figure 2A*). Further, DHFR levels could be lowered in the *mgrB*-knockout by deletion of *phoP* or *phoQ*, but not *rpoS* or *iraM* (*Figure 2A*). The increase in DHFR levels was due to greater activity of the *folA* promoter in *E. coli ΔmgrB* compared to wild-type (*Figure 2B and C*), and this too was *phoP*-dependent but *rpoS*-independent (*Figure 2C*). Importantly, the *folA* promoter retained trimethoprim-stimulation in the *mgrB*-knockout (*Figure 2C*). The *folA* promoter is not a known target for PhoP, and does not harbor sequences similar to the PhoP binding site. Consequently, we were unable to detect direct binding of PhoP to the *folA* promoter in vitro by gel shift assays (*Figure 2— figure supplement 2*). Thus, the enhancement of *folA* expression by PhoP was most probably an indirect effect.

In line with higher expression of DHFR in *mgrB*-deficient *E. coli*, 9 of the 10 trimethoprim-resistant isolates (TMPR1–2, 4–10) showed elevated DHFR protein levels (*Figure 2D*). The only isolate that did not (TMPR3), harbored the W30G mutation in *folA*, which we have shown earlier to result in lower steady-state levels of DHFR (*Matange et al., 2018*; *Matange, 2020*). Further, deletion of *phoP* resulted in lower expression of DHFR in TMPR1–5 (*Figure 2E*). These data demonstrated that loss of *mgrB* conferred trimethoprim tolerance in *E. coli* by enhancing DHFR protein levels, through PhoP-dependent, RpoS-independent upregulation of the *folA* promoter.

## Derepression of PhoPQ facilitates resistance evolution by altering the fitness landscape of *folA* mutations

In addition to *mgrB* mutations, isolates TMPR1, 3, and 5 harbored mutations in *folA* (*Table 1*). Likewise, TMPR4 harbored a large genomic duplication encompassing the *folA* gene along with other genes that may confer trimethoprim resistance (*Table 1*). To assess the contribution of loss of *mgrB* to the phenotypes of these isolates, we tested trimethoprim MIC and IC50 of *ΔphoP* derivatives of TMPR1–5. Remarkably, loss of *phoP* greatly reduced the MIC of trimethoprim for all five isolates (*Figure 3A*). TMPR2 and 4 that did not harbor mutations in *folA* were completely re-sensitized to trimethoprim by *phoP* deletion (*Figure 3A and B*). TMPR1, 3, and 5, which harbored mutations in *folA* retained some trimethoprim-resistance after deletion of *phoP*, but with close to tenfold reduction in drug IC50 (*Figure 3B*). Thus, even though loss of *mgrB* itself did not appreciably alter drug MIC, it significantly potentiated the phenotypes of *folA* mutations in trimethoprim-resistant *E. coli*.

The above result prompted us to ask whether mutation at *mgrB* influenced the selection dynamics of trimethoprim resistance in *E. coli*. The genetic constitution of antibiotic-resistant bacteria is known to be influenced by drug concentration (*Matange et al., 2019*; *Wistrand-Yuen et al., 2018*; *Zhou et al., 2000*). Hence, we mapped the fitness landscape of drug-tolerant (*E. coli ΔmgrB*) and drug-resistant (TMPR1, 3, and 5) strains across different concentrations of trimethoprim (*Figure 3C*). Each of the above strains was allowed to compete against a GFP-expressing derivative of *E. coli* wild-type (*E. coli*-GFP). Since GFP fluorescence of the mixed culture would indicate the fractional abundance of *E. coli*-GFP in the population, we used it as a read-out of relative fitness of the test strains (designated as competitive quotient; a value of 0 indicated no change in fitness relative to wild-type, >0 signified higher relative fitness, <0 signified reduced relative fitness). Over the entire range of trimethoprim concentrations used by us, TMPR1, 3, and 5 performed better than *ΔmgrB* alone. This difference was most pronounced at concentrations that approached the MIC value for the wild-type. Next, we asked how *ΔphoP* derivatives of TMPR1, 3, and 5 would perform in a similar assay (*Figure 3D*). Interestingly, *ΔmgrB* was fitter than *ΔphoP* derivatives of TMPR1, 3, and 5 across all concentrations of trimethoprim tested by us. These results explained why *mgrB* mutations were the predominant early adaptive event in our selection experiments and indicated that resistance-conferring DHFR mutations were likely to be selected only after PhoPQ signaling had been derepressed.

The above experiments showed that mutations in *folA* and *mgrB* had a synergistic effect on the phenotype of trimethoprim-resistant *E. coli*. We had shown earlier that several mutations in DHFR, notably those at Trp30, are detrimental for its stability and result in aggregation, proteolysis, and reduced levels (*Matange et al., 2018*). Therefore, we hypothesized that enhanced expression due to PhoPQ derepression could compensate for loss of mutant DHFRs due to misfolding. To test this, we expressed wild-type DHFR or its W30G/W30R mutant alleles from an IPTG-inducible promoter and checked the impact that expression level had on drug IC50 (*Figure 3E*). While trimethoprim IC50 was enhanced by ~3 -fold upon overproduction of wild-type DHFR, it was potentiated by ×16- and ×8-fold for W30G and W30R DHFR alleles, respectively. We performed the same assay with W30F and W30L mutant alleles of DHFR. These alleles have comparable in vivo sensitivity to trimethoprim, but only W30L destabilizes the protein (*Matange et al., 2018*). Overexpression affected the phenotype of W30F to a similar extent as wild-type DHFR, while the IC50 of trimethoprim for *E. coli* expressing W30L DHFR was stimulated ×8-fold upon overexpression (*Figure 3E*). Thus, mutant DHFR alleles resulted in higher gains in drug IC50 upon overexpression, explaining the synergy between *mgrB* and *folA* mutations, and this was related to their destabilizing effects on DHFR.

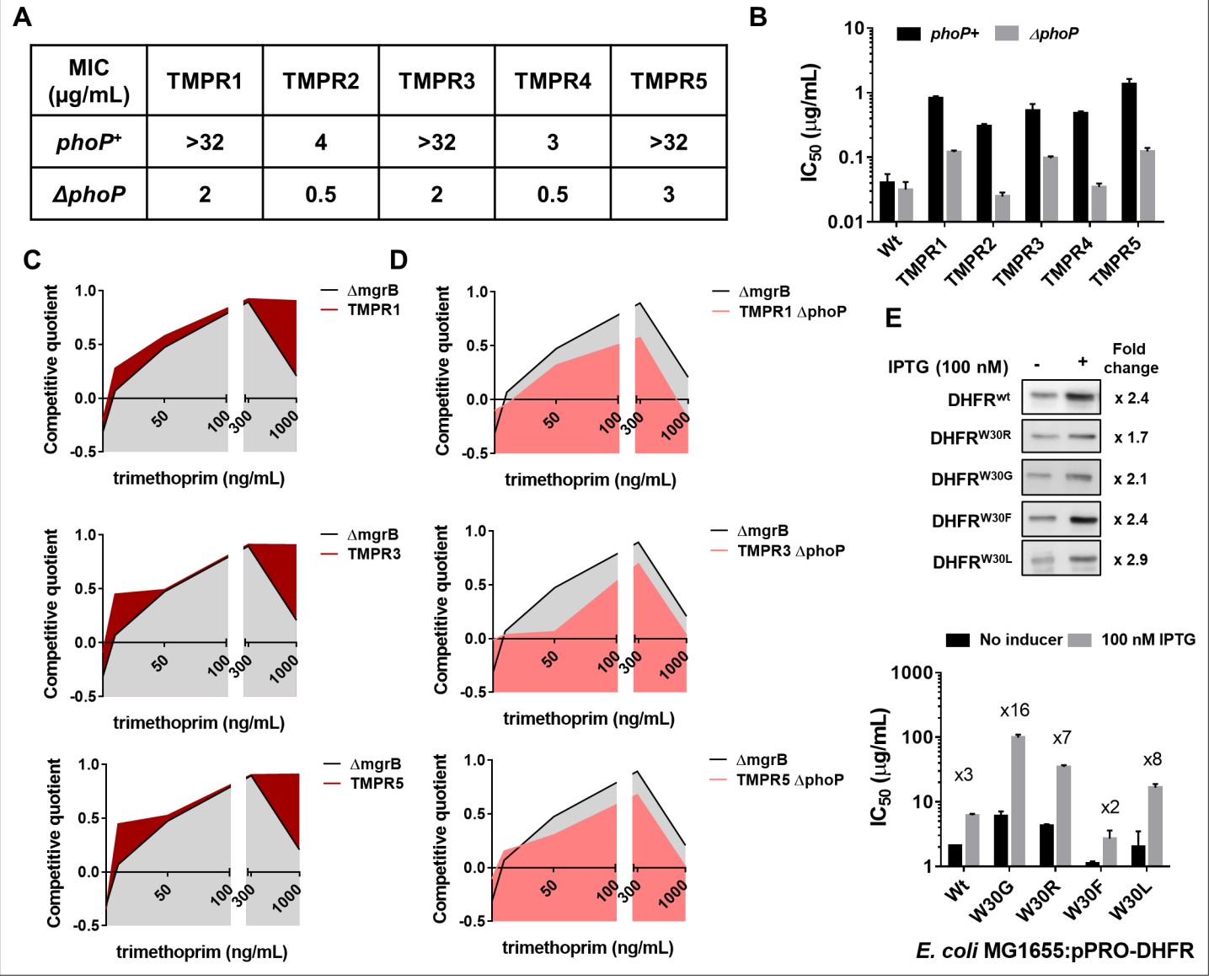

**Figure 3.** Synergy between *mgrB* and *folA* mutations alters the fitness landscape of trimethoprim resistant *Escherichia coli*. (A) MIC values of trimethoprim for resistant isolates (TMPR1–5) and their *ΔphoP* derivatives calculated from E-tests. (B) IC$_{50}$ values of trimethoprim from resistant isolates (TMPR1–5) and their *ΔphoP* derivatives. Mean± SEM from three independent experiments is plotted. (C, D) Fitness landscape of trimethoprim resistant isolates TMPR1, 3, and 5 (C) or their *ΔphoP* derivatives (D) compared with trimethoprim tolerant *E. coli ΔmgrB*. Competitive quotients were calculated by competition with *E. coli*-GFP (wild-type *E. coli* expressing GFP) in varying concentrations of trimethoprim. Higher values indicate higher fitness relative to wild-type. Representative data from three replicates are shown. (E) Effect of overexpression of DHFR on IC$_{50}$ of trimethoprim monitored by expression of wild-type or mutant DHFR from an IPTG-inducible promoter. Upper panel. Immunoblots showing overexpression of all DHFR alleles tested. The fold increase in DHFR expression level upon addition of IPTG was calculated based on band intensities. Lower panel. IC$_{50}$ values of trimethoprim for various mutants with and without inducer are plotted (mean± SD from three replicates). The fold increase in IC$_{50}$ in the presence of inducer is indicated for each mutant. DHFR, dihydrofolate reductase; MIC, minimum inhibitory concentration.

## Cost compensation drives evolution of the PhoPQ transcriptional network during long-term antibiotic exposure

Having established a new mechanism for trimethoprim tolerance, and its effect as a potentiator of trimethoprim resistance, we next sought to understand the long-term evolutionary consequences of this adaptation. We established three lineages of *E. coli* evolving in high concentration of trimethoprim (300 ng/ml, ×0.5-fold MIC) for ~350 generations (designated WTMP300 A, B, and C). In agreement with results from short-term trimethoprim exposures (*Figure 1A*, *Table 1*), mutations in the

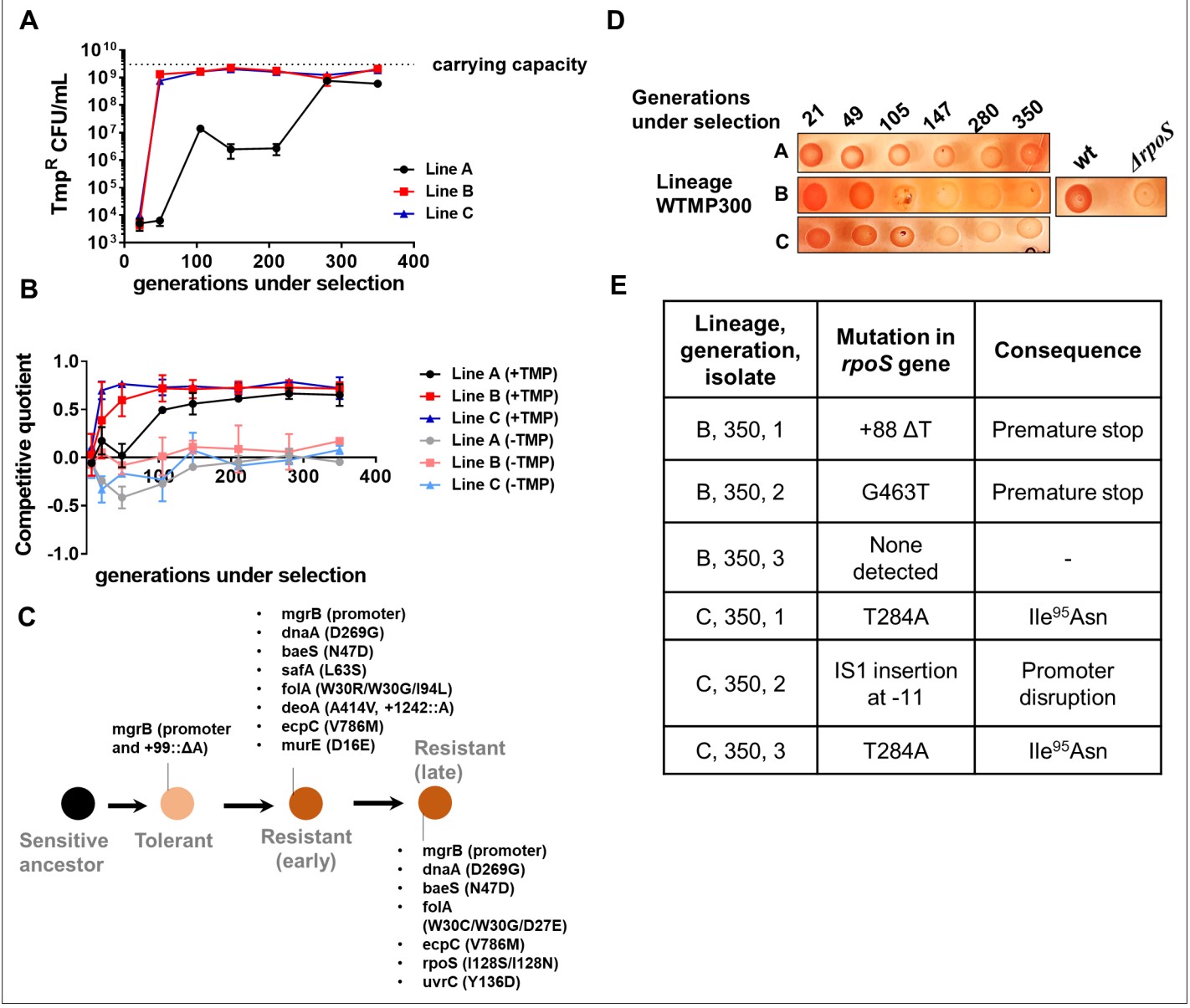

**Figure 4.** Genomic changes in *Escherichia coli* adapting to high trimethoprim concentrations. (**A**) Titre of trimethoprim-resistant bacteria in WTMP300 lineages A, B and C during the course of evolution. Each data point represents mean± SD from two to three measurements. The typical carrying capacity of *E. coli* growing in drug-free medium (2–3×10⁹ CFU/ml) is indicated with a dotted line. (**B**) Competitive quotient of WTMP300 lineages (**A–C**) in drug-free or trimethoprim supplemented media was monitored by competition with *E. coli*-GFP (wild-type *E. coli* expressing GFP). Higher values of competitive quotient indicate higher fitness relative to the wild-type ancestor. Trimethoprim was used at 300 ng/ml. Each point represents mean± SD from two to three measurements. (**C**) Schematic representation of genomic changes associated with early and late adaptation to trimethoprim in WTMP300 lineage A (see *Supplementary file 1* for complete list). (**D**) Congo red staining of WTMP300 lineages A, B, and C to verify loss of active RpoS. Controls (wild-type and *E. coli ΔrpoS*) are shown for reference. Representative data from three replicates are shown. (**E**) Sanger sequencing of resistant isolates from lineages B and C confirming the accumulation of mutations in the *rpoS* gene.

The online version of this article includes the following figure supplement(s) for figure 4:

**Figure supplement 1.** IC₅₀ values of trimethoprim for resistant isolates from WTMP300 lineages at 47, 157, and 350 generations.

*mgrB* gene or its promoter swept through all three lineages within the first ~20 generations of antibiotic exposure (*Supplementary file 1*). Subsequently, at this high drug pressure, there was rapid enrichment of resistant bacteria that were eventually fixed in all three lineages (*Figure 4A*).

Next, we tracked adaptation of the evolving lineages to trimethoprim by competing them against *E. coli*-GFP. For all three lineages, competitive quotient rose steeply initially and then plateaued

(*Figure 4B*). The initial rapid phase of adaptation was concordant with enrichment of trimethoprim-resistant bacteria (*Figure 4A and B*). Interestingly, IC50 of trimethoprim for resistant isolates from all three lineages plateaued early during evolution and did not change significantly over time (*Figure 4—figure supplement 1*). Thus, once resistance had evolved, its fixation in the population did not require further gains in drug MIC/IC50. On the other hand, competitive quotients of evolving populations in trimethoprim-free media initially decreased (*Figure 4B*), most likely due to the costs of mutations at the *mgrB* locus (*Figure 1D*). However, all three lineages recovered over time such that at 350 generations of evolution their relative fitness was similar to or slightly better than the ancestor (*Figure 4B*). Taken together, we interpreted these results to mean that trimethoprim-resistant bacteria enhanced their fitness over long-term evolution by amelioration of fitness costs, rather than enhancement in drug IC50.

To understand the genetic basis of these adaptations, we sequenced the genomes of up to five resistant isolates from lineage A at three different time points, that is, 105, 147, and 350 generations of evolution (*Supplementary file 1*, *Figure 4C*). This lineage was chosen for genetic analyses, since it showed the slowest accumulation of resistant bacteria, providing an opportunity to capture the sequence of mutations accumulated during resistance evolution. Sequenced isolates from all three time points had mutations at the *mgrB* locus, re-iterating its role as a driver of early adaptation to trimethoprim (*Supplementary file 1*, *Figure 4C*). Mutations at the *folA* locus were also common among isolates at all three time points, explaining the high IC50 of trimethoprim for these isolates (*Supplementary file 1*, *Figure 4C*). In addition, some isolates harbored mutations at the *baeS* locus, a known regulator of efflux pump expression (*Nagakubo et al., 2002*), and *safA*, a regulator of PhoPQ signaling (*Eguchi et al., 2011*; *Eguchi et al., 2007*; *Supplementary file 1*, *Figure 4C*). Strikingly, all sequenced isolates from 350 generations, but not earlier time points, harbored substitution of Ile128 to Ser or Asn in RpoS (*Supplementary file 1*, *Figure 4C*). The Ile128 residue of the RpoS sigma factor is critical for its activity, and hydrophilic substitutions at this site inactivate it (*Iwase et al., 2017*). In order to confirm loss of RpoS activity in these lineages, we exploited the differential staining of RpoS-deficient and RpoS-expressing *E. coli* by Congo red. Congo red stains curli fibers and cellulose, which are produced only by bacteria with active RpoS (*Smith et al., 2017*). In agreement with results from genome-sequencing, all three lineages showed progressively poorer staining with Congo red (*Figure 4D*). We also PCR amplified and Sanger sequenced the *rpoS* gene from three trimethoprim isolates each derived from lineages B and C at 350 generations. Five of the six isolates sequenced harbored mutations at the *rpoS* locus (*Figure 4E*). Thus, while resistance-associated genetic changes were observed early during evolution, loss-of-function mutations in *rpoS* were associated with adaptation during long-term exposure to trimethoprim.

Loss of RpoS, we had found, did not itself confer trimethoprim resistance (*Figure 1C*). Further, since RpoS is stimulated by PhoPQ signaling, it was reasonable to expect that loss of RpoS activity compensated for the costs of *mgrB*-mutations. In support of this suggestion, RpoS-deficient bacteria were selected over RpoS-expressing bacteria only in a Δ*mgrB* background (*Figure 5A*). Moreover, this effect was relatively independent of the presence of trimethoprim (*Figure 5A*). RpoS activates the transcription of a number of stress-responsive and stationary phase genes. Several of these genes are upregulated in an *mgrB*-knockout (*Xu et al., 2019*). Could the costs of *mgrB* deletion be related to unrequired expression of stress and stationary phase genes? We tested this hypothesis by generating knockouts of four RpoS-regulated genes, that is, *gadX*, *cbpA*, *fic,* and *sra* that are upregulated in a *mgrB*-deficient background (*Xu et al., 2019*). Knockouts of three of these, that is, *gadX*, *cbpA*, and *fic* were all selected over *E. coli* Δ*mgrB* independent of the presence of trimethoprim (*Figure 5A*). These data confirmed that loss of RpoS compensated for the fitness costs of PhoPQ dereperssion by ameliorating the expression of unwanted stationary-phase and stress-related genes (*Figure 5B*).

## Strength of antibiotic selection determines the choice of adaptive strategy during the evolution of the PhoPQ-folA-RpoS transcriptional network

Our experiments so far identified two possible strategies available to *E. coli* for enhancing fitness subsequent to evolving tolerance, that is, resistance and cost compensation. We next asked what motivated the "choice" of strategy that an evolving bacterial population would adopt. Cost compensation is known to allow resistant bacteria to enhance their fitness in drug-free or low antibiotic

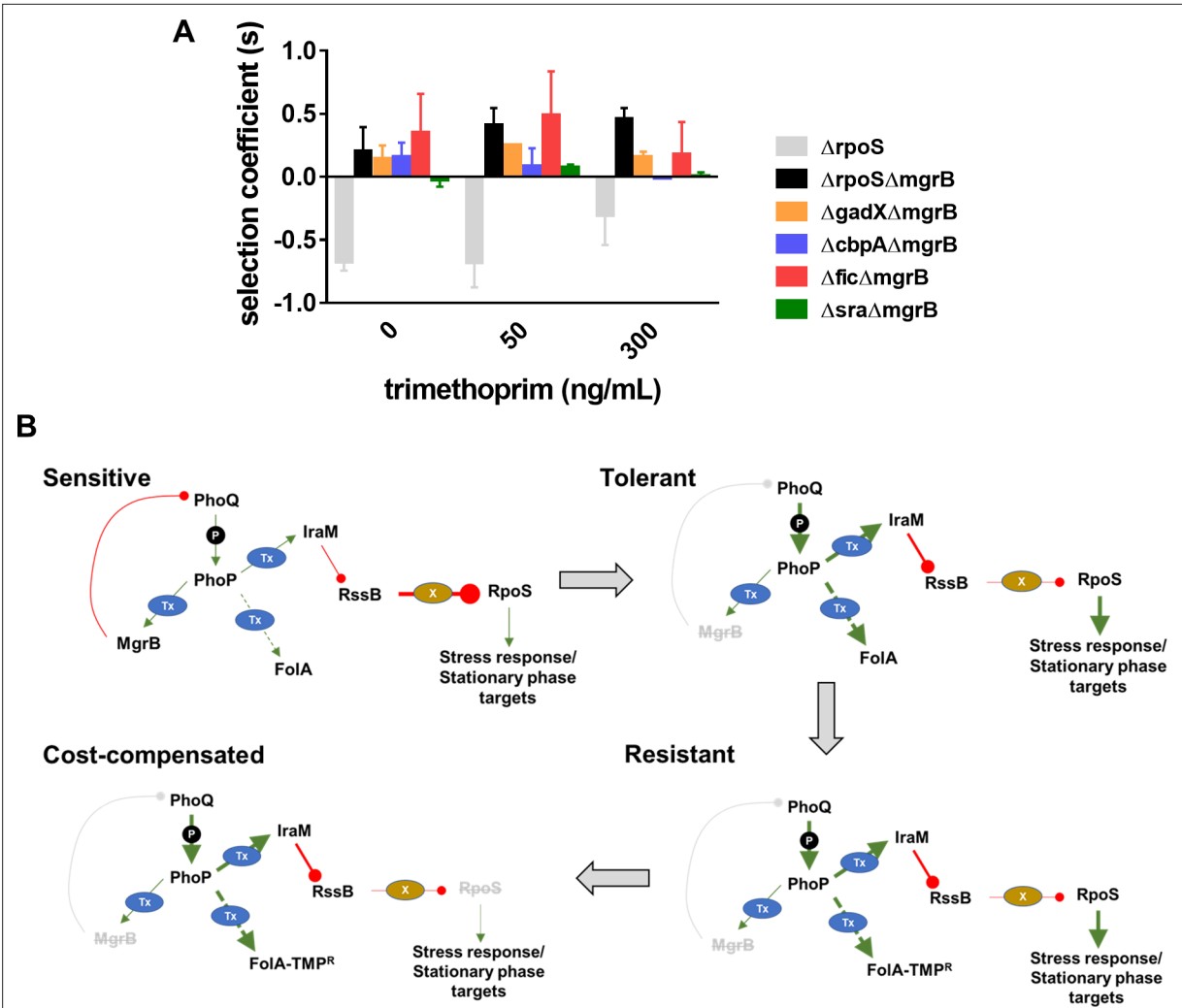

**Figure 5.** Compensation of fitness costs of *mgrB*-deficiency by loss of RpoS. (**A**) Selection coefficient of *Escherichia coli ΔrpoS* relative to wild-type (gray bars), *E. coli ΔmgrBΔrpoS* relative to *E. coli ΔmgrB* (black bars), or indicated mutants in the *ΔmgrB* background relative to *E. coli ΔmgrB* (colored bars). Appropriate strains were allowed to compete in the absence or presence of trimethoprim at the indicated concentrations for ~9 generations. Data represent mean± SEM from three independent measurements. (**B**) Schematic representation of adaptation of *E. coli* to high trimethoprim over long-term evolution. Genetic changes and mechanisms associated with sensitive, tolerant, resistant, and cost-compensated phenotypes are shown. The evolutionary sequence is indicated by the direction of arrows. Activating interactions are shown by green arrows, while inhibitory interactions are shown by red lines. Strength of each interaction is qualitatively represented by the thickness of arrows. Indirect interactions are shown as discontinuous lines. Phosphorylation is indicated by 'P.' Transcriptional changes are indicated by 'Tx.' Proteolytic degradation is indicated by 'X.' Inactivating mutations are represented by gray text with strikethrough.

conditions. Therefore, we argued that antibiotic concentration may determine which adaptive strategy was favored. In order to test this, we mixed tolerant (*E. coli ΔmgrB*), cost-compensated tolerant (*E. coli ΔmgrBΔrpoS*), and resistant (TMPR1) strains (initial mixing ratio 98:1:1) and allowed them to compete for ~9 generations in different concentrations of trimethoprim. In the absence of trimethoprim or in low concentrations of the antibiotic, *ΔmgrBΔrpoS* was strongly enriched over *ΔmgrB* and TMPR1 (*Figure 6A*). However, at higher drug concentrations, TMPR1 was favored over its competitors (*Figure 6A*). This clear dependency on trimethoprim concentration suggested that at low drug pressures, evolutionary outcomes were likely to be biased in favor of cost compensation rather than resistance-evolution.

We empirically tested the above prediction in three replicate lineages of *E. coli* K-12 MG1655 evolving under low trimethoprim pressure (50 ng/ml, 0.08 ×MIC, designated WTMP50 A, B, and C). In support of our hypothesis, resistant bacteria did not exceed more than 1 % of the population in

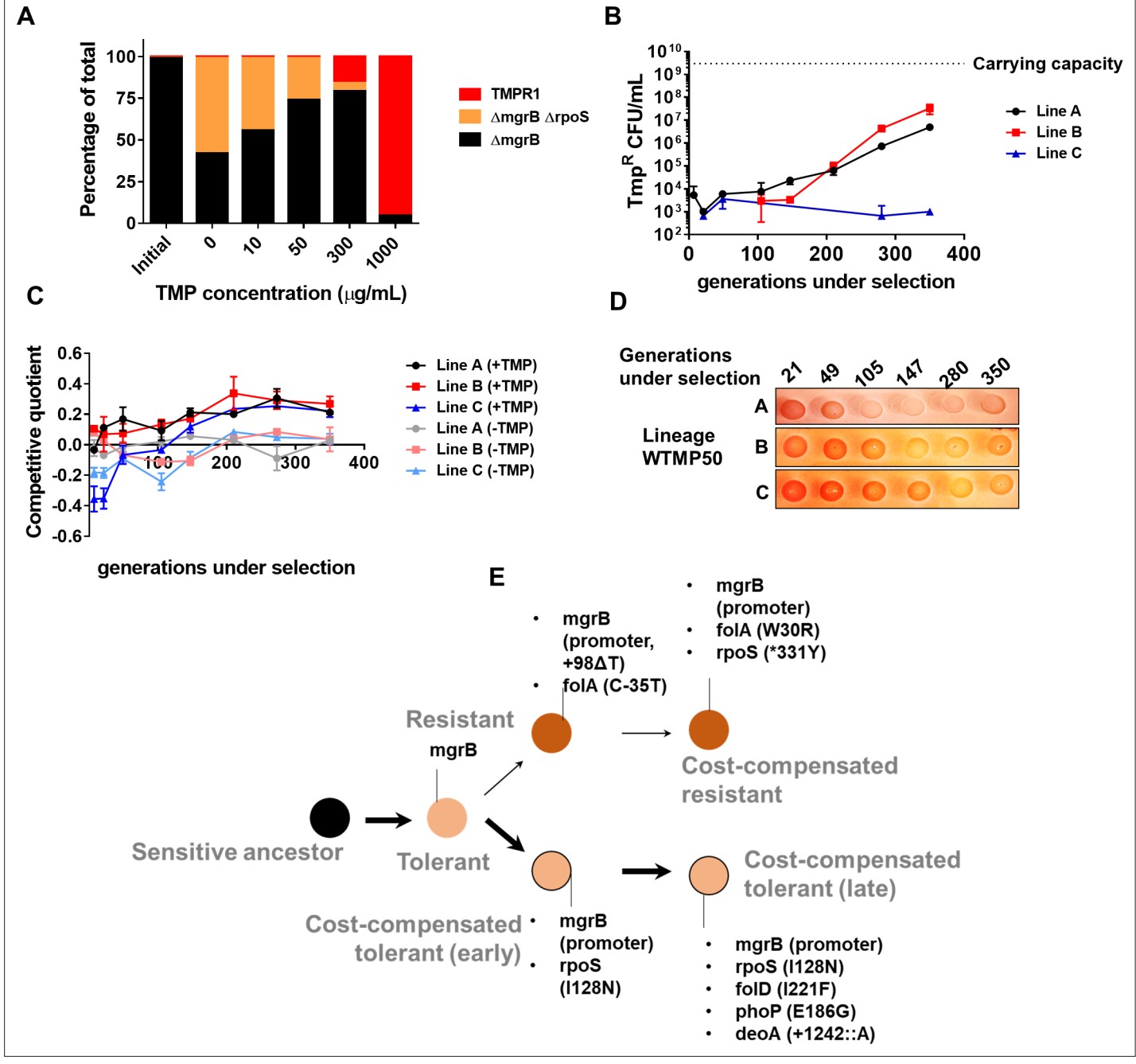

**Figure 6.** Genomic changes in *Escherichia coli* adapting to low trimethoprim concentrations. (**A**) Three-strain competition between *E. coli ΔmgrB* (tolerant), *E. coli ΔmgrBΔrpoS* (cost-compensated tolerant) and *E. coli* TMPR1 (resistant). Initial mixing ratio of strains was 98:1:1, respectively. Ratios of strains after ~9 generations of competition at the indicated concentration of trimethoprim are shown. Data are representative of three replicates. (**B**) Titre of trimethoprim-resistant bacteria in WTMP50 lineages A, B, and C during the course of evolution. Each data point represents mean± SD from two to three measurements. The typical carrying capacity of *E. coli* growing in drug-free medium (2–3×10⁹ CFU/ml) is indicated with a dotted line. (**C**) Competitive quotient of WTMP50 lineages A, B, and C in drug-free or trimethoprim supplemented media was monitored by competition with *E. coli-*GFP (wild-type *E. coli* expressing GFP). Higher values of competitive quotient indicate higher fitness relative to the wild-type ancestor. Trimethoprim was used at 50 ng/ml. Each point represents mean± SD from two to three measurements. (**D**) Congo red staining of WTMP50 lineages A, B, and C to verify loss of active RpoS in all lines. Representative data from three replicates are shown. (**E**) Schematic representation of genomic changes associated with early and late adaptation to trimethoprim in WTMP50 lineage A (see ***Supplementary file 2*** for complete list). Two parallel lineages branching out from tolerant bacteria in the WTMP50 lines are shown.

these lineages for the duration of the experiment (*Figure 6B*). Indeed, in WTMP50 lineage C, the fraction of resistant bacteria stayed below 0.001 % of the total population throughout the experiment (*Figure 6B*). Gain in relative fitness of WTMP50 lineages over the ancestor was lower than WTMP300, and showed very poor concordance with enrichment of trimethoprim-resistant bacteria over time (*Figure 6C*). Importantly, like WTMP300, all three WTMP50 lineages got fitter in antibiotic-free medium over time (*Figure 6C*) and lost RpoS activity (*Figure 6D*). These data suggested that there were at least two parallel evolving adaptive trajectories in the WTMP50 lines, that is, resistant bacteria, that made up a minority of the population and cost-compensated, tolerant bacteria that made up the majority of the population. Genome sequencing of resistant and tolerant isolates from 147 to 350 generations under selection from WTMP50 lineage A, confirmed that this was indeed the case (*Supplementary file 2*, *Figure 6E*). The genetic trajectory of resistant bacteria in WTMP50 was very similar to the WTMP300, in that mutations in *folA* were acquired before cost-compensatory mutations in *rpoS* (*Supplementary file 2*, *Figure 6E*). On the other hand, tolerant bacteria acquired *rpoS* mutations earlier during evolution and did not acquire mutations in folA even after long-term

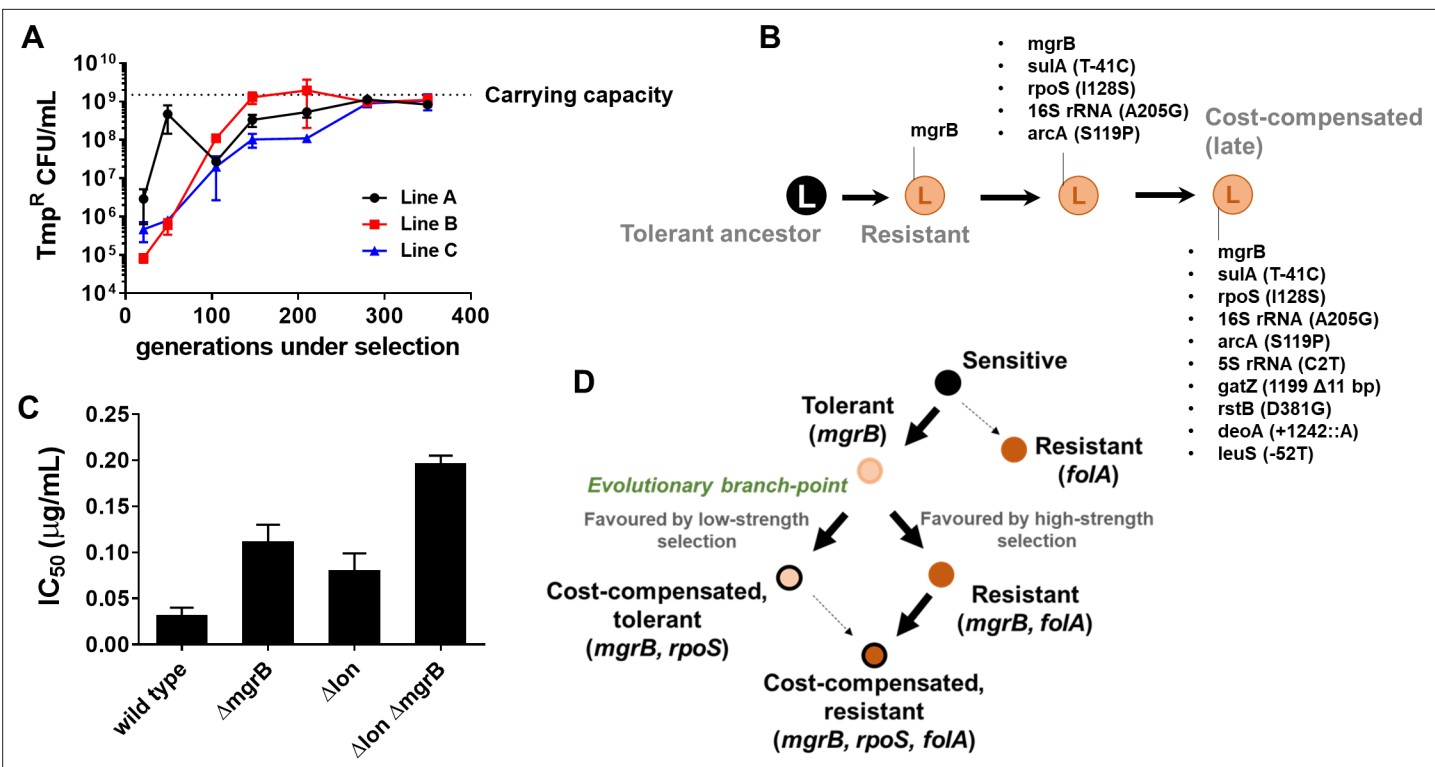

**Figure 7.** Genomic changes in *Escherichia coli* Δ*lon* adapting to high trimethoprim concentrations. (**A**) Titre of trimethoprim-resistant bacteria in LTMP300 lineages A, B, and C during the course of evolution. Each data point represents mean± SD from two to three measurements. The typical carrying capacity of *E. coli* Δ*lon* growing in drug-free medium (1–2×10⁹ CFU/ml) is indicated with a dotted line. (**B**) Schematic representation of genomic changes associated with early and late adaptation to trimethoprim in LTMP300 lineage A (see *Supplementary file 3* for complete list). (**C**) Additive effect of Δ*lon* and Δ*mgrB* mutations on IC$_{50}$ of trimethoprim for *E. coli*. Mean± SD values of IC$_{50}$ from three independent experiments are plotted. (**D**) Model of adaptation in the PhoPQ-folA-RpoS axis showing dependency of genetic changes and associated phenotypes on drug pressure. Implicated genetic loci are indicated in parentheses. Evolutionary transitions identified in this study are shown as solid black arrows. Possible transitions that are unlikely to occur based on the results from this study are shown as dotted arrows.

The online version of this article includes the following figure supplement(s) for figure 7:

**Figure supplement 1.** Predicted protein sequences of mutant MgrB alleles identified in this study, and from publicly available sequences of pathogenic and environmental *Escherichia coli* strains identified by BLAST analysis. Each sequence is colored based on the functional annotation of MgrB. Accession numbers and source of strain (*Souvorov et al., 2018*; *Tyson et al., 2019*) are indicated in the table.

**Figure supplement 2.** Predicted protein sequences of MgrB alleles from publicly available sequences of pathogenic and environmental *Salmonella typhimurium* and *Citrobacter freundii* strains identified by BLAST analysis. Each sequence is colored based on the functional annotation of MgrB for *Escherichia coli*. Accession numbers and source of strain are indicated in the table.

antibiotic exposure (*Supplementary file 2*, *Figure 6E*). Thus, cost compensation was indeed the preferred strategy for adaptation at lower antibiotic concentrations.

Drug concentrations are generally equated with strength of selection for antibiotic resistance (*Andersson and Hughes, 2014*; *Gullberg et al., 2011*). However, selection strength is determined by intrinsic factors as well, such as the genetic background (*Matange, 2020*; *Matange et al., 2019*). Therefore, we tested whether the link between selection strength and evolutionary trajectory would hold true in another genetic background. We have shown earlier that Lon protease-deficient *E. coli* are trimethoprim-tolerant due to a *folA*-independent, acrAB efflux pump-dependent mechanism (*Matange, 2020*). For *E. coli Δlon*, it is expected that 300 ng/ml of trimethoprim would represent a milder selection strength than wild-type. Therefore, we established three lineages of *E. coli Δlon* that experienced trimethoprim (300 ng/ml, 0.5 ×MIC, designated LTMP300 A, B, and C) for 350 generations and examined their adaptation over time. At the phenotypic level, LTMP300 lineages resembled WTMP300 in terms of fixation of resistant-bacteria (*Figure 7A*). However, at the genotypic level there was substantial similarity between LTMP300 and WTMP50 lineages. We found no *folA* mutations in resistant isolates from the LTMP300 lineage A (*Supplementary file 3*, *Figure 7B*). Instead, trimethoprim-resistant bacteria derived from an early time point (147 generations) had already accumulated inactivating mutations in *rpoS* in addition to mutations in *mgrB* (*Supplementary file 3*, *Figure 7B*). Thus, while the LTMP300 lineages had the genetic signature of evolution at low selection strength, they phenotypically resembled high selection strength. This discrepancy was explained by the fact that *Δlon* and *ΔmgrB* showed an additive effect on trimethoprim IC50 (*Figure 7C*). While loss of *mgrB* alone resulted in drug tolerance, together with Lon-deficiency it resulted in drug resistance. Importantly, once *mgrB* mutations were acquired, LTMP300 lineages evolved by compensatory mutations rather than resistance-conferring mutations in *folA*, similar to WTMP50. These results, therefore, re-iterated the role of selection strength in determining the genetic trajectory followed by *E. coli* during adaptation to trimethoprim (*Figure 7D*).

## Discussion

The PhoPQ signaling cascade is a well-studied two-component system, not just in Enterobacteria but also other species. Traditionally thought to be regulated by $Mg^{2+}$ (*Groisman, 2001*), this system also responds to low pH (*Eguchi et al., 2011*) and periplasmic redox state (*Lippa and Goulian, 2012*). Mounting evidence in recent years has thrown light on its role in resistance to antibiotics, particularly antimicrobial peptides such as polymyxins (*Mmatli et al., 2020*). The clinical relevance of these findings was brought to the fore when a number of studies identified inactivating mutations in *mgrB* in carbapenem/colistin-resistant *Klebsiella pneumoniae* (*Cannatelli et al., 2013*; *Haeili et al., 2017*; *Silva et al., 2021*). Derepression of PhoPQ in *Klebsiella* results in alterations in the structure of Lipid A, conferring resistance to colistin (*Kidd et al., 2017*). Inactivation of *mgrB* also collaterally enhances virulence in *Klebsiella* (*Kidd et al., 2017*), a particularly worrying observation given that polymyxins represent the last line of defense against hospital-borne Gram-negative infections at the moment. Similarly in *E. coli*, C18G, another antimicrobial peptide, leads to growth arrest and filamentation by activating PhoPQ signaling (*Yadavalli et al., 2016*).

In the context of trimethoprim resistance, mutations in *mgrB* were reported earlier by *Baym et al., 2016* from drug-resistant *E. coli* evolving on mega petri-plates (*Baym et al., 2016*), though a mechanistic basis for this change was not known. Transcriptional upregulation of *folA* upon loss of *mgrB*, a main result from our study, fills this lacuna. Additionally, by showing the synergistic interaction between mutations in *mgrB* and *folA*, our study also establishes how loss of *mgrB* facilitates resistance evolution. These results are in line with other studies that have identified the facilitatory role that drug tolerance plays in the evolution of resistance (*Levin-Reisman et al., 2017*).

Early inactivation of *mgrB* in *E. coli* upon trimethoprim exposure raises two questions. First, how does PhoPQ enhance the expression of *folA*? The only known transcriptional regulator of *folA* in *E. coli* is the TyrR transcription factor (*Yang et al., 2007*). However, we were unable to detect any change in the expression level of DHFR protein in a *tyrR* deletion strain (*Figure 2—figure supplement 3*). Given the widespread gene regulatory impact of PhoPQ derepression in *E. coli*, including its effects on other transcription factors, its effects on *folA* levels are likely to be secondary or tertiary. These indirect regulatory effects of PhoPQ derepression were sufficient to drive the selection of *mgrB* mutations. Further investigation into the mechanism of *folA* promoter upregulation in *mgrB*-deficient *E.*

*coli* is likely to reveal additional regulators for this housekeeping gene. It is important to note that a larger number of loss-of-function mutations at *mgrB* are possible than gain-of-function mutations at the *folA* locus. This may also be an important factor explaining why *mgrB* mutations occurred earlier in our experiments than *folA* mutations.

Second, is inactivation of MgrB relevant in clinical or environmental strains of *E. coli*? We detected four kinds of mutations at the *mgrB* locus in our experiments, the most frequent being disruption of the promoter of *mgrB*. Additionally, we found three different single nucleotide insertions or deletions within the coding region. These mutations are expected to result in a longer or shorter protein with markedly altered C-terminal sequence as a result of translational frame-shift (*Figure 7—figure supplement 1*). We searched for protein sequences similar to these frame-shifted alleles of MgrB in the non-redundant protein sequence database. BLAST analysis revealed nine clinical and environmental *E. coli* strains from four different pathogen surveillance and sequencing projects that had altered C-terminal sequences similar to the mutant alleles found by us (*Figure 7—figure supplement 1*). Indeed, one of these strains, *E. coli* NCTC9075 from the collection of Public Health England, harbored exactly the same deletion in *mgrB* (+100 ΔA) that we identified in some of our laboratory-evolved strains (*Figure 7—figure supplement 1*). We also identified mutant MgrB sequences from three isolates of *Salmonella enterica* and one isolate of *Citrobacter freundii* (*Figure 7—figure supplement 2*). Thus, the relevance of MgrB in trimethoprim tolerance, or indeed in other phenotypes of pathogenic Enterobacteria, is likely to be far more significant that currently understood.

Though we have focussed on the compensatory effects of RpoS mutations in this study, genomics analyses revealed a number of other loci that accumulated mutations in our laboratory evolution experiments. For instance, mutations in the s. For instance, mutations in the *deoA* locus, predicted to cause premature termination of the DeoA protein, were detected in resistant isolates from all selection schemes employed by us (*Table 1*, *Supplementary files 1–3*). DeoA, a thymidine phosphorylase, is part of the pyrimidine catabolism pathway in *E. coli* (*Razzell and Casshyap, 1964*). Deactivation of this pathway is known to be beneficial for *E. coli* harboring hypomorphic DHFR alleles by reducing the demand for de novo pyrimidine biosynthesis (*Rodrigues and Shakhnovich, 2019*). Interestingly, mutations in some of the other loci from our study showed strict dependency on drug concentration used during selection. Most notably, mutations in *baeS*, *ecpC*, and *dnaA* genes, which code for two-component kinase, fimbrial chaperone, and replication initiator proteins respectively, were found in a majority of resistant isolates from the WTMP300 lineages, but not in WTMP50 lineages (*Supplementary files 1 and 2*). BaeS, sensor kinase of the BaeRS two-component system, has been associated with multi-drug resistance in a number of bacteria including *E. coli* due to its effect on the expression of efflux pumps (*Nagakubo et al., 2002*). The fimbrial chaperone EcpC, on the other hand, has never been associated with resistance to antibiotics, and is potentially a novel locus worth investigating. Point mutations in DnaA (D269G and D280G) that were detected in trimethoprim-resistant isolates from WTMP300 lineages are expected to compromise its ATP-binding (*Erzberger et al., 2006*). Indeed, D269 lies in the 'Sensor I' motif of DnaA and its mutation to Ala reduces ATPase activity of the protein in vitro (*Kawakami et al., 2006*). We envisage two possible reasons for these mutations in DnaA to be selected during adaptation to trimethoprim. First, thymine-less death, induced by thymine starvation or trimethoprim treatment, can be prevented in *E. coli* by lowered rates of replication initiation (*Guzmán and Martín, 2015*). Mutations in DnaA that compromise its ATPase activity would help achieve this. Second, DnaA mutations in clinical and laboratory strains of *M. tuberculosis* have been associated with isoniazid resistance due to altered gene expression profiles (*Hicks et al., 2020*), and a similar mechanism may be operative in the case of *E. coli* and trimethoprim. The appearance of mutations at these loci only at high antibiotic pressure is likely to be a result of the higher pressure for resistance.

We also observed mutations specific to genetic background. Most notable of these were mutations in 5 S and 16 S ribosomal RNAs, promoter of *sulA* and the ArcB transcriptional regulator in trimethoprim-resistant isolates evolved from the *Δlon* strain. The Lon protease degrades a number of housekeeping proteins, but can also degrade misfolded mutant proteins (*Tsilibaris et al., 2006*). We have shown in the past that a Lon-deficient background permits access to a larger repertoire of mutations to adapting bacterial populations (*Matange, 2020*). This phenomenon may very well explain why certain mutations were detected only in the Lon-deficient background. Further, mutations in the *sulA* gene promoter are expected to prevent its expression (*Cole, 1983*), thus alleviating the

accumulation of this cell division inhibitor in the Lon mutant. Loss of *sulA* is advantageous for *E. coli* Δ*lon* as it rescues the hyper-filamentation defect of this strain (*Matange, 2020*; *Gottesman et al., 1981*), which may explain why mutations in the *sulA* promoter were identified only in the LTMPR300 lineages.

To our knowledge, the present study is one of a kind in being able to trace the evolutionary trajectory of a gene regulatory network from its initial perturbation in response to selection, to eventual evolutionary rescue by compensatory mutations. Three main observations from this study significantly contribute to a general understanding of how gene regulatory networks change in response to selection. First, changes in gene regulatory networks can alter the fitness landscape of adaptive mutations. The evolutionary potential of gene regulatory changes is well-recognized particularly in the context of vertebrate development (*Spitz and Duboule, 2008*). Similarly, gene regulatory changes as mediators of evolution have been suggested in a series of synthetic biology experiments in prokaryotes (*Bayer, 2010*). Our study demonstrates that gene regulatory mutations can also act as facilitators of subsequent evolutionary change by shifting the fitness landscape of mutant alleles. This finding adds a new dimension to understanding the role of gene regulatory networks in mediating the evolution of new functions. Further, mutant alleles of enzymes, like in the case of DHFR, are known to often display trade-offs with stability (*Matange et al., 2018*; *Rodrigues et al., 2016*). The role of gene regulatory networks in negotiating these trade-offs needs to be factored in to understanding the genetics of evolutionary adaptation.

Second, cost compensation emerged as an important force driving the evolution of the PhoPQ-folA-RpoS network. Evolution of gene regulatory networks is thought to be primarily under stabilizing selection, in light of empirically observed properties such as developmental canalization and mutational robustness (*Gilad et al., 2006*). Compensatory evolution in regulatory networks has been understood mainly in the context of cis or trans-regulatory changes that minimize variation in gene expression (*Signor and Nuzhdin, 2018*; *Thompson et al., 2015*). Our results demonstrate that the fitness effects of adaptive mutations can be significant. As a result, compensation of fitness costs can also drive additional changes in regulatory networks. Importantly, loss of RpoS activity, a key cost-compensatory change in our experiments, is a commonly encountered trait in Enterobacteria (*Dong et al., 2009*; *Kotewicz et al., 2003*; *Martinez-Garcia et al., 2003*). RpoS mutants are associated with fitness advantage in aging bacterial colonies (*Wrande et al., 2008*), and natural isolates of *E. coli* often have inactive alleles of this sigma factor (*Dong et al., 2009*; *Kotewicz et al., 2003*; *Martinez-Garcia et al., 2003*; *Chiang et al., 2011*; *Liu et al., 2017*). The driving force behind loss of RpoS in Enterobacteria is thought to be selected for enhanced growth rate (*Dong et al., 2009*; *Liu et al., 2017*). Since RpoS competes with housekeeping sigma factor RpoD for RNA polymerase occupancy, loss of RpoS drives growth and replication, and compromises stress tolerance as a trade-off (*Dong et al., 2009*; *Liu et al., 2017*). The results of our study support this thesis. However, in our study, selection of RpoS mutants was driven in response to gene regulatory perturbation and not by environmental factors. Given the high prevalence of RpoS-mutations in natural isolates of Enterobacteria, it is very likely that they may serve as compensatory changes for other gene regulatory perturbations as well.

Finally, the dependency of evolutionary strategy on selection pressure is a key finding from this study (*Figure 6D*). Over the last decade, there has been significant attention given to how sub-lethal drug concentrations, such as those found in natural reservoirs of bacteria, alter the evolution of drug resistance (*Matange et al., 2019*; *Wistrand-Yuen et al., 2018*; *Zhou et al., 2000*; *Andersson and Hughes, 2014*; *Baquero, 2001*; *Gutierrez et al., 2013*; *Hiltunen et al., 2017*). The present study provides direct evidence linking drug concentration to the genetics of adaptation to antibiotics in bacteria. We also provide a mechanistic understanding for this link. Cost compensation by second-site mutations is well-known for antimicrobial resistance and is indeed a widely-accepted phenomenon (*Björkman et al., 2000*; *Handel et al., 2006*; *Melnyk et al., 2015*; *Qi et al., 2016*; *Schulz zur Wiesch et al., 2010*). It is thought to occur under relaxed selection, that is, when drug is not present in the environment. Our study demonstrates that compensation of cost and resistance evolution can also be thought of alternative strategies for enhancing fitness in the presence of the antibiotic. The relative gain in fitness from each of these strategies is determined not just by drug concentration but also by genetic background, and both these factors have vital influence on the evolutionary outcomes of bacteria under antibiotic selection.

## Materials and methods

### Strains, plasmids, and culture conditions

*E. coli* K-12 MG1655 or its derivates were cultured in Luria-Bertani broth supplemented with appropriate antibiotics at 37 °C with shaking (180–200 rpm). The various strains and plasmids used in this study and their sources are listed in *Table 2*. Single gene knockouts were obtained from the Keio collection (*Baba et al., 2006*; *Yamamoto et al., 2009*) (from the NBRP Resource, National Institute of Genetics, Japan) and moved into the MG1655 background using P1 transduction. For double gene knockouts, antibiotic marker cassette was first removed using Flp recombinase (pCP20-flp plasmid) (*Datsenko and Wanner, 2000*), and then subjected to P1 transduction.

Antibiotics used in this study were purchased from Himedia (India) or Sigma-Merck (USA, Germany). Kanamycin (30 µg/ml), chloramphenicol (25 µg/ml), or ampicillin (100 µg/ml) were added to liquid or semi-solid media just before inoculation as required. For isolating trimethoprim-resistant mutants from laboratory evolution experiments Luria-Bertani agar was supplemented with 1 µg/ml trimethoprim.

### Antibiotic sensitivity

Trimethoprim sensitivity of gene knockouts or evolved clones was tested using the following methods:

Minimum inhibitory concentration (MIC): MIC determination was performed using E-test. Trimethoprim strips (x–x µg/ml) were procured from Himedia (India) and used as per the manufacturer's instruction.

Inhibitory concentration-50 (IC50): IC50 of trimethoprim was determined using a broth dilution assay as described in *Matange et al., 2019* and *Matange et al., 2018*.

Colony formation assay: Colony-forming efficiency of wild-type or mutant *E. coli* was calculated by spotting 10 µl of neat or serially diluted (tenfold dilution series) late log-phase cultures on Luria-Bertani agar supplemented with 0, 50, 100, 300, 500, and 1000 ng/ml trimethoprim. Plates were incubated for 18 hr at 37 °C before counting colonies.

Colonization efficiency: Wild-type or mutant *E. coli* cultures were grown to saturation and then serially diluted. Increasing bacterial CFUs (~$10^4$, $10^5$, $10^6$, and $10^7$) were added to 1 ml of Luria-Bertani broth supplemented with 0, 500, 1000, or 5000 ng/ml trimethoprim. Cultures were grown for 18 hr at 37 °C and optical density (at 600 nm) was measured in a spectrophotometer.

### Relative fitness and competitive growth assays

Relative fitness of gene knockouts or evolved clones was estimated using competition assays. Test strains were mixed with ancestral *E. coli* from saturated mono-cultures (1:1, unless otherwise mentioned) in 1–3 ml Luria-Bertani broth supplemented with trimethoprim at the appropriate concentration. Strains were allowed to compete for ~9 generations.

For 2-strain or 3-strain competition, CFU/ml for each strain was determined by plating serially diluted cultures on antibiotic-free (total CFU) and kanamycin (knock-out strains) or trimethoprim (trimethoprim resistant isolates) supplemented Luria-Bertani agar. Relative fitness (w) or selection coefficient (s) was calculated using the following formulae:

$$w = ln\left(\frac{Tf}{Ti}\right)/ln\left(\frac{Rf}{Ri}\right)$$

$$s = 1 - w$$

where Tf and Ti are the final and initial CFU/ml of the test strain and Rf and Ri are the final and initial CFU/ml of the reference strain, respectively.

For competitive quotients, GFP-expressing derivative of *E. coli* wild-type (*E. coli*-GFP) was used as the reference strain. At the end of the competition experiment, 200 µl of each mixed culture was aliquoted into a 96-well plate and GFP fluorescence and optical density (at 600 nm) were estimated in a multi-mode plate reader (Varioskan-Thermo Scientific or Enlight-PerkinElmer). For each individual experiment, two controls were maintained, that is, *E. coli* -GFP alone and control competition between *E. coli*-GFP and wild-type. GFP fluorescence was normalized to optical density (nGFP) and used to calculate competitive quotient using the following formula:

**Table 2.** List of strains and plasmids used in this study.

| Strain | Description | Source |
|---|---|---|
| *Escherichia coli* wild-type | *E. coli* K-12 MG1655 | – |
| TMPR1–10 | Trimethoprim resistant derivatives of *E. coli* wild-type obtained by short term laboratory selection with 300 ng/ml of trimethoprim | *Matange et al., 2018* |
| TMPR1–5 *ΔphoP::Kan* | Derivatives of TMPR1–5 in which *phoP* is replaced with a Kanamycin resistance cassette | This study. Mutation moved from BW25113 background (Keio collection) to TMPR1–5 by P1 transduction |
| *E. coli ΔmgrB::Kan* | Derivative of *E. coli* wild-type in which *mgrB* is replaced with a Kanamycin resistance cassette | This study. Mutation moved from BW25113 background (Keio collection) to MG1655 by P1 transduction |
| *E. coli ΔmgrB (unmarked)* | Derivative of *E. coli ΔmgrB::Kan* without Kanamycin resistance marker | This study. Kanamycin cassette removed using Flp-recombinase |
| *E. coli ΔphoP::Kan* | Derivative of *E. coli* wild-type in which *phoP* is replaced with a Kanamycin resistance cassette | This study. Mutation moved from BW25113 background (Keio collection) to MG1655 by P1 transduction |
| *E. coli ΔmgrB ΔphoP::Kan* | Derivative of *E. coli ΔmgrB (unmarked)* in which *phoP* is replaced with a Kanamycin resistance cassette | This study. Mutation moved from BW25113 background (Keio collection) to MG1655 by P1 transduction |
| *E. coli ΔphoQ::Kan* | Derivative of *E. coli* wild-type in which *phoQ* is replaced with a Kanamycin resistance cassette | This study. Mutation moved from BW25113 background (Keio collection) to MG1655 by P1 transduction |
| *E. coli ΔmgrB ΔphoQ::Kan* | Derivative of *E. coli ΔmgrB (unmarked)* in which *phoQ* is replaced with a Kanamycin resistance cassette | This study. Mutation moved from BW25113 background (Keio collection) to MG1655 by P1 transduction |
| *E. coli ΔiraM::Kan* | Derivative of *E. coli* wild-type in which *iraM* is replaced with a Kanamycin resistance cassette | This study. Mutation moved from BW25113 background (Keio collection) to MG1655 by P1 transduction |
| *E. coli ΔmgrB ΔiraM::Kan* | Derivative of *E. coli ΔmgrB (unmarked)* in which *iraM* is replaced with a Kanamycin resistance cassette | This study. Mutation moved from BW25113 background (Keio collection) to MG1655 by P1 transduction |
| *E. coli ΔrpoS::Kan* | Derivative of *E. coli* wild-type in which *rpoS* is replaced with a Kanamycin resistance cassette | This study. Mutation moved from BW25113 background (Keio collection) to MG1655 by P1 transduction |
| *E. coli ΔmgrB ΔrpoS::Kan* | Derivative of *E. coli ΔmgrB (unmarked)* in which *rpoS* is replaced with a Kanamycin resistance cassette | This study. Mutation moved from BW25113 background (Keio collection) to MG1655 by P1 transduction |
| *E. coli-GFP* | Derivative of *E. coli* wild-type in which gene coding for GFP is integrated into the genome downstream of the *aidB* locus | Kind gift from Dr. Amrita Hazra (Indian Institute of Science Education and Research, Pune, India) |
| *E. coli ΔgadX::Kan* | Derivative of *E. coli* wild-type in which *gadX* is replaced with a Kanamycin resistance cassette | This study. Mutation moved from BW25113 background (Keio collection) to MG1655 by P1 transduction |

*Table 2 continued on next page*

*Table 2 continued*

| Strain | Description | Source |
|---|---|---|
| *E. coli ΔmgrB ΔgadX::Kan* | Derivative of *E. coli ΔmgrB (unmarked)* in which *gadX* is replaced with a Kanamycin resistance cassette | This study. Mutation moved from BW25113 background (Keio collection) to MG1655 by P1 transduction |
| *E. coli ΔcbpA::Kan* | Derivative of *E. coli* wild-type in which *cpbA* is replaced with a Kanamycin resistance cassette | This study. Mutation moved from BW25113 background (Keio collection) to MG1655 by P1 transduction |
| *E. coli ΔmgrB ΔcbpA::Kan* | Derivative of *E. coli ΔmgrB (unmarked)* in which *cbpA* is replaced with a Kanamycin resistance cassette | This study. Mutation moved from BW25113 background (Keio collection) to MG1655 by P1 transduction |
| *E. coli Δsra::Kan* | Derivative of *E. coli* wild-type in which *sra* is replaced with a Kanamycin resistance cassette | This study. Mutation moved from BW25113 background (Keio collection) to MG1655 by P1 transduction |
| *E. coli ΔmgrB Δsra::Kan* | Derivative of *E. coli ΔmgrB (unmarked)* in which *sra* is replaced with a Kanamycin resistance cassette | This study. Mutation moved from BW25113 background (Keio collection) to MG1655 by P1 transduction |
| *E. coli Δfic::Kan* | Derivative of *E. coli* wild-type in which *fic* is replaced with a Kanamycin resistance cassette | This study. Mutation moved from BW25113 background (Keio collection) to MG1655 by P1 transduction |
| *E. coli ΔmgrB Δfic::Kan* | Derivative of *E. coli ΔmgrB (unmarked)* in which *fic* is replaced with a Kanamycin resistance cassette | This study. Mutation moved from BW25113 background (Keio collection) to MG1655 by P1 transduction |
| *E. coli Δlon::Kan* | Derivative of *E. coli* wild-type in which *lon* is replaced with a Kanamycin resistance cassette | *Matange, 2020* |
| *E. coli Δlon::Cmp* | Derivative of *E. coli* wild-type in which *lon* is replaced with a Chloramphenicol resistance cassette | *Matange, 2020* |
| *E. coli Δlon::Cmp ΔmgrB::Kan* | Derivative of *E. coli ΔmgrB::Kan* in which *lon* is replaced with a Chloramphenicol resistance cassette | This study. Mutation moved from *E. coli Δlon::Cmp* to *E. coli ΔmgrB::Kan* by P1 transduction |
| *E. coli ΔtyrR::Kan* | Derivative of *E. coli* wild-type in which *tyrR* is replaced with a Kanamycin resistance cassette | This study. Mutation moved from BW25113 background (Keio collection) to MG1655 by P1 transduction |
| *E. coli ΔmgrB ΔtyrR::Kan* | Derivative of *E. coli ΔmgrB (unmarked)* in which *tyrR* is replaced with a Kanamycin resistance cassette | This study. Mutation moved from BW25113 background (Keio collection) to MG1655 by P1 transduction |
| **Plasmid** | **Description** | **Source** |
| pCP20-flp | Temperature sensitive plasmid for expression of Flp recombinase | Coli Genetic Stock Center (CGSC), Yale University, USA |
| pUA66-PfolA-GFP | Plasmid harboring GFP under the *folA* gene promoter. Reporter plasmid for activity of P*folA*. | Kind gift from Dr. Sanchari Bhattachrya (Harvard University, USA) |
| pPRO-DHFR wild-type/ W30G/W30R/W30F/W30L | Plasmids for overexpression of HexaHis-DHFR and its mutants. DHFR is downstream of an IPTG-inducible promoter. | *Matange et al., 2018* |

*Table 2 continued on next page*

*Table 2 continued*

| Strain | Description | Source |
|---|---|---|
| pCA24N-PhoP | Plasmid for overexpression of HexaHis-PhoP | ASKA(−) collection (**Kitagawa et al., 2005**), NBRP Resource, National Institute of Genetics, Japan |

$$competitive\,quotient = ([nGFP(c) - nGFP(T)] - [nGFP(c) - nGFP(wt)]) \div ([nGFP(c) - nGFP(wt)])$$

where nGFP(c), nGFP(T), and nGFP(wt) are the normalized GFP fluorescence values of *E. coli*-GFP alone, mixed culture of *E. coli* -GFP, and test strain, and mixed culture of *E. coli*-GFP and wild-type, respectively.

## Immunoblotting

Immunoblotting was used to determine the expression level of DHFR in mutants or resistant isolates as described in *Matange, 2020*. FtsZ was used as a loading control. Anti-DHFR IgG was used at a concentration of 100 ng/ml. Anti-FtsZ (kind gift from Prof. Manjula Reddy, CCMB, India) polyclonal antiserum was used at a dilution of 1:50,000. Band intensities were quantitated in ImageJ software.

## Activity of *folA* gene promoter

The activity of the promoter of *folA* was measured in wild-type or mutant *E. coli* using the pUA66-P$_{folA}$-GFP reporter plasmid (kind gift from Dr. Sanchari Bhattacharya, Department of Chemistry & Chemical Biology, Harvard University). Plasmid reporter was transformed into appropriate strains and GFP fluorescence was measured over growth in a multi-mode plate reader (Varioskan-Thermo Scientific). For comparison across strains, GFP fluorescence was normalized to optical density.

## Purification of His-tagged PhoP

Hexa-His tagged PhoP was overexpressed in *E. coli* K-12 MG1655 and purified using Ni-NTA affinity chromatography. Bacterial strain harboring pCA24N-PhoP plasmid was grown in 50–100 ml Luria-Bertani broth till an optical density of ~0.8. Protein production was induced using IPTG (500 µM). Induction was carried out for 3 hr at 37 °C, following which bacterial cultures were centrifuged at 3000 ×*g* for 10 min. Bacterial pellets were stored at –80 °C until needed. Cells were lysed by sonication in Lysis buffer (Tris-HCl pH 8.0 100 mM, NaCl 100 mM, β-mercaptoethanol 1 mM, and glycerol 10%) and loaded onto a Ni-NTA-agarose resin (Bio-Rad, USA). The resin was washed using Wash buffer (Tris-HCl pH 8.0 100 mM, NaCl 300 mM, β-mercaptoethanol 1 mM, and imidazole 20 mM) and eluted using Elution buffer (Tris-HCl pH 8.0 100 mM, NaCl 100 mM, β-mercaptoethanol 1 mM, imidazole 300 mM, and glycerol 10%). Eluted protein was desalted using PD-10 desalting columns (Sigma-Merck, USA/Germany). Concentration of the desalted protein was estimated using Bradford's method and the protein was stored at –80 °C until needed.

## Electrophoretic mobility shift assay

For electrophoretic mobility shift assay (EMSA), biotinylated probes shown in *Figure 2—figure supplement 2* were synthesized (Sigma-Aldrich, USA) and annealed to a final concentration of 25 pmol/µl. They were further diluted to 25 fmol/µl in nuclease-free water. Recombinant purified PhoP protein was phosphorylated in vitro by incubating with 20 mM acetyl-phosphate and 10 mM MgCl$_2$ for 45 min at 37 °C. The phosphorylated protein was used immediately for further experiments. For binding and visualization, LightShift Chemilunescent EMSA Kit (Thermo Fisher Scientific, USA) was used as per the manufacturer's instructions.

## Laboratory evolution of trimethoprim resistance

For long-term laboratory evolution of trimethoprim resistance, replicate lineages (three for each drug concentration/genotype) were established using serial passaging in trimethoprim-supplemented Luria-Bertani broth (2 ml). Bacteria (1 %) were transferred to fresh media every 10–14 hr such that each growth cycle had between six and seven generations. Laboratory evolution was continued for ~350

generations (i.e., 50 growth cycles). Periodically an aliquot of culture from each lineage was mixed with an equal volume of sterile glycerol (50%) and frozen at –80 °C for analysis.

The titer of trimethoprim-resistant bacteria from each of the frozen stocks was determined by spotting 10 µl of serially diluted culture on Luria-Bertani agar supplemented with trimethoprim (1 µg/ml). Resistant colonies (3–6) were selected at random for MIC determination or sequencing. Isolation of tolerant bacteria from WTMP50 lineages was done by spotting serially diluted cultures on drug-free medium. Tolerant isolates (3–6) were re-inoculated and their MIC was determined to confirm that they were not resistant to trimethoprim.

## Congo red staining of bacterial colonies

Congo red staining was used to determine RpoS activity in evolved lineages. Saturated cultures (10 µl) of appropriate strains were spotted on petri-plates containing YESCA medium (0.1% w/v yeast extract, 1% w/v casein hydrolysate, and 2% w/v agar) supplemented with Congo red (50 µg/ml) and Coomassie Brilliant Blue G-250 (10 µg/ml). Plates were incubated at 28 °C to induce Curli fibers and photographed after ~24 hr of growth.

## Sequencing and genome analysis

Genomic DNA was extracted from evolved resistant/tolerant isolates using phenol:chloroform:iso-amyl alcohol extraction as described in *Matange et al., 2019*. Extracted gDNA was then cleaned up using QiaAMP DNA purification spin columns (QIAGEN, Germany) and quantitated spectrophotometrically. Integrity of the prepared gDNA was verified by gel electrophoresis before further applications.

The *folA* locus (encompassing the coding region and 195 bp upstream promoter) was PCR amplified from gDNA of trimethoprim-resistant isolates as described in *Matange et al., 2018* and Sanger sequenced. The mgrB and rpoS loci were similarly PCR amplified using the mgrB_seq_fwd (5′- CAAC CTCTTCTCTTTTTATGTTCGC-3′) and mgrB_seq_rev (5′-CAACCAAAGACGCAATGTTCATCACG-3′) primers or rpoS_seq_fwd (5′-CGGAACCAGGCTTTTGCTTGAATG-3′) and rpoS_seq_rev (5′-GGCC AGCCTCGCTTGAGACTGGCC-3′) primers respectively and Sanger sequenced.

For whole-genome resequencing, paired-end whole-genome next-generation sequencing (NGS) was performed on a MiSeq system (Illumina, USA) with read lengths of 150 bps. Processed reads were aligned to the reference genome *E. coli* K-12 MG1655 (NZ_CP025268) using bowtie2. A cutoff of a minimum of 20 × coverage was employed, which made >97% of the reads for each of the samples permissible for analysis. Variant calling and prediction of new junctions in the genome to identify large structural mutations, such as insertions and deletions, were done using Breseq using default settings (*Deatherage and Barrick, 2014*). All variants and new junctions that were already present in the wild-type ancestor were excluded.

Sanger sequencing services were provided by FirstBase (Malaysia). Library preparation and NGS services were provided by Bencos (India).

## Acknowledgements

The authors would like to acknowledge Dr. Amrita Hazra (IISER, Pune, India) and Dr. Manjula Reddy (CCMB, Hyderabad, India) for strains and reagents. The authors would like to acknowledge National Bioresource, NIG, Japan, for *E. coli* mutants. This project was funded by the INSPIRE program (Department of Science and Technology, Govt. of India) and IISER-Pune. NM is a recipient of the SERB-Research Scientist Award (Science Education and Research Board, Govt. of India) and the DBT-Wellcome India Alliance Intermediate Fellowship.

## Additional information

### Funding

| Funder | Grant reference number | Author |
|---|---|---|
| Department of Science and Technology, Ministry of Science and Technology, India | | Nishad Matange |
| Science and Engineering Research Board | | Nishad Matange |

The funders had no role in study design, data collection and interpretation, or the decision to submit the work for publication.

### Author contributions

Vishwa Patel, Project administration; Nishad Matange, Conceptualization, Data curation, Formal analysis, Funding acquisition, Investigation, Methodology, Project administration, Supervision, Writing - original draft, Writing - review and editing

### Author ORCIDs

Nishad Matange (iD) http://orcid.org/0000-0002-2947-3931

### Decision letter and Author response

Decision letter https://doi.org/10.7554/eLife.70931.sa1
Author response https://doi.org/10.7554/eLife.70931.sa2

## Additional files

### Supplementary files

• Supplementary file 1. Summary of genetic changes in trimethoprim resistant isolates from WTMP300 Lineage A identified by genome sequencing.

• Supplementary file 2. Summary of genetic changes in trimethoprim resistant and tolerant isolates from WTMP50 Lineage A by genome sequencing.

• Supplementary file 3. Summary of genetic changes in trimethoprim resistant isolates from LTMP300 Lineage A by genome sequencing.

• Transparent reporting form

• Source data 1. Source data for *Figure 2A,D,E*; *Figure 3E* and *Figure 2—figure supplement 3*. *Figure 2*. Uncropped, annotated immunoblot (Source data 1) and raw image files (Source data 2, 3) for DHFR and FtsZ in various *E. coli* mutants. Specific bands were identified based on molecular weight and absence from lysate from *E. coli* ΔfolA. *Figure 2D*. Uncropped, annotated immunoblot (Source data 1) and raw image files (Source data 2, 3, 4) for DHFR and FtsZ in various trimethoprim resistant *E. coli* mutants. Specific bands were identified based on molecular weight and absence from lysate from *E. coli* ΔfolA. *Figure 2E*. Uncropped, annotated immunoblot (Source data 1) and raw image files (Source data 2, 3, 4) for DHFR and FtsZ in various trimethoprim resistant *E. coli* mutants (TMPR1-5) and their ΔphoP derivatives. Specific bands were identified based on molecular weight and absence from lysate from *E. coli* ΔfolA. *Figure 3E*. Uncropped, annotated immunoblot (Source data 1) and raw image files (Source data 2, 3) for plasmid-expressed His-tagged DHFR or its mutant alleles in *E. coli* in the presence of indicated concentrations of inducer (IPTG). *Figure 2—figure supplement 3*. Uncropped, annotated immunoblot (Source data 1) and raw image files (Source data 2, 3) for DHFR and FtsZ in indicated *E. coli* mutants. Specific bands were identified based on molecular weight and absence from lysate from *E. coli* ΔfolA.

### Data availability

Raw data from whole genome sequencing experiments used for this study have been deposited to GenBank under Project ID PRJNA741586.

The following dataset was generated:

| Author(s) | Year | Dataset title | Dataset URL | Database and Identifier |
|---|---|---|---|---|
| Patel V, Matange N | 2021 | Trimethoprim resistance in E. coli | https://www.ncbi.nlm.nih.gov/bioproject/PRJNA741586 | NCBI BioProject, PRJNA741586 |

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
