## [Decision Letter]

**Acceptance summary:**

This paper investigates the evolutionary path of *Escherichia coli* resistance to the antibiotic trimethoprim. The authors show that adaptive mutations that accumulate early are often not in the drug target itself, but rather mutations that lead to transcriptional up-regulation of the drug target. Higher-level resistance can then evolve due to the addition of mutations in the drug target; however, at lower drug concentrations, cells are more likely to accumulate mutations that reverse the fitness defect associated with the initially acquired mutations. Overall, this study shows that regulatory mutations can play a major role in the evolution of antibiotic resistance in bacterial populations, and that the evolutionary path is influenced by the level of drug exposure.

**Decision letter after peer review:**

Thank you for submitting your article "Adaptation and Compensation in a Bacterial Gene Regulatory Network Evolving Under Antibiotic Selection" for consideration by *eLife*. Your article has been reviewed by 3 peer reviewers, including Joseph T Wade as the Reviewing Editor and Reviewer #1, and the evaluation has been overseen by Patricia Wittkopp as the Senior Editor. The following individuals have agreed to reveal their identity: Srujana Samhita Yadavalli (Reviewer #2); Daria Van Tyne (Reviewer #3).

Essential revisions:

The reviewers were generally supportive of the manuscript, with only minor suggestions. There is one essential revision that requires an experiment: we request that you sequence mgrB and surrounding sequence for the 5 mutants where the specific mutations were not identified (TMPR6-10). Based on the data presented, a mutation in mgrB seems likely for these strains. For cases where the mutation is not in mgrB, whole-genome sequencing for one or more of the strains would be appropriate, since it would be useful to know of mgrB-independent mutations that lead to phoP-dependent up-regulation of DHFR.

The reviewers also felt that the paper was difficult to follow in a few places, and two of the reviewers have specific suggestions to help improve the flow.

*Reviewer #1 (Recommendations for the authors):*

Figure 1D. The fitness numbers shown at the bottom of the figure panel are confusing. I suggest moving these to the main text.

Figure 1D. The different lines are hard to tell apart from one another. The use of different colors can help here.

Figure 1E. The MIC assay is qualitative, and doesn't agree well with the other assays of drug sensitivity. I don't think it is reasonable to conclude that the MIC is the same for the wt and mgrB mutant strains.

*Reviewer #2 (Recommendations for the authors):*

-Please check and fix the entire manuscript including figures, axes, legends for typological or grammatical errors.

-Insert missing references as indicated in the review.

*Reviewer #3 (Recommendations for the authors):*

I believe the manuscript would be improved by addressing the following comments:

1. The term "tolerance" should be clearly defined, and if possible the authors should point to prior literature supporting the use of this term for the phenomena observed here. I agree that increased IC50 without increased MIC is clearly not resistance, but the term "tolerance" is often employed to describe decreased rates of killing for bacteriocidal antibiotics. Since trimethoprim is bacteriostatic, I'm not sure "tolerance" should be used over another term like "decreased susceptibility."

2. The authors only sequenced half of their evolved isolates (Figure 1), however in Figure 2D they show DHFR protein abundance for all 10 selected lines. They suggest that loss of mgrB confers trimethoprim tolerance by enhancing DHFR protein levels, however mgrB status (wild type versus mutant) should be established for all 10 lines. This would be simple to do with PCR and Sanger sequencing if needed (PCR alone would be sufficient to identify IS element insertions upstream of mgrB). Showing phenotype (i.e. protein abundance) data for all 10 lines but only genotype data for half of them, when it is very easy to genotype the other half, feels incomplete.

3. Similarly, the authors should confirm rpoS mutations in lineages B/C in Figure 4 and Figure 6. Doing so would be easy with PCR and Sanger sequencing, and would further support their conclusions.

4. In panel 3B, the δ-phoP strain in a wild type background should be included as well, to remind readers that phoP deletion alone does not affect TMP MIC.

5. In Figure 4, the trajectory of lineage A looks different from the other two lineages. The authors should make a note of this, and provide a justification for focusing on genetically characterizing this lineage rather than the other two.

6. In Line 397, the authors should briefly explain the folA-independent mechanism of TMP tolerance in the lon-deficient *E. coli* strain.

---

## [Author Response]

Essential revisions:The reviewers were generally supportive of the manuscript, with only minor suggestions. There is one essential revision that requires an experiment: we request that you sequence mgrB and surrounding sequence for the 5 mutants where the specific mutations were not identified (TMPR6-10). Based on the data presented, a mutation in mgrB seems likely for these strains. For cases where the mutation is not in mgrB, whole-genome sequencing for one or more of the strains would be appropriate, since it would be useful to know of mgrB-independent mutations that lead to phoP-dependent up-regulation of DHFR.

We have now amplified and Sanger sequenced the mgrB gene and its promoter from all 10 TMPR isolates. As expected, we do indeed find mutations at the mgrB promoter in all 10 isolates. These data have been added to the revised manuscript in the Figure 1A schematic.

The reviewers also felt that the paper was difficult to follow in a few places, and two of the reviewers have specific suggestions to help improve the flow.

We thank the Reviewers for bringing this to our notice and have made changes as suggested. We hope that the revised manuscript is easier to read.

Reviewer #1 (Recommendations for the authors):Figure 1D. The fitness numbers shown at the bottom of the figure panel are confusing. I suggest moving these to the main text.

We have modified Figure 1D as suggested and moved values of relative fitness to the main text (Page 5, Line 149-151).

Figure 1D. The different lines are hard to tell apart from one another. The use of different colors can help here.

We have incorporated colours for the different lines in Figure 1D as suggested by Reviewer 1.

Figure 1E. The MIC assay is qualitative, and doesn't agree well with the other assays of drug sensitivity. I don't think it is reasonable to conclude that the MIC is the same for the wt and mgrB mutant strains.

We agree that the E-strip assay alone is qualitative. However, we think that the conclusion that wt and mgrB mutant have a similar MIC is justified, since we obtained similar values in broth dilution experiments as well (Figure 1B). Therefore, we have not modified this conclusion in the revised version of the manuscript.

Reviewer #2 (Recommendations for the authors):-Please check and fix the entire manuscript including figures, axes, legends for typological or grammatical errors.

We thank the Reviewer for bringing these errors to our notice. We have made the necessary corrections.

-Insert missing references as indicated in the review.

We have inserted the missing reference in the revised version of the manuscript.

Reviewer #3 (Recommendations for the authors):I believe the manuscript would be improved by addressing the following comments:1. The term "tolerance" should be clearly defined, and if possible the authors should point to prior literature supporting the use of this term for the phenomena observed here. I agree that increased IC50 without increased MIC is clearly not resistance, but the term "tolerance" is often employed to describe decreased rates of killing for bacteriocidal antibiotics. Since trimethoprim is bacteriostatic, I'm not sure "tolerance" should be used over another term like "decreased susceptibility."

We thank Reviewer 3 for this important comment. We struggled significantly with how to best define/designate/name the phenotype of the mgrB mutant. As pointed out by the Reviewer, tolerance has been used in recent studies in the context of killing rate of bacteria upon exposure to high concentrations of bactericidal antibiotics. However, for a bacteriostatic antibiotic the term remains ill-defined. We have chosen to retain the term ‘tolerance’ in our manuscript due to the following parallels between tolerance for bactericidal antibiotics and mgrB-trimethoprim:

i. Both phenomena are genetic in origin.

ii. Both phenomena result in enhanced survival/growth in drug-supplemented media but unaltered drug MICs.

iii. Both phenomena are precursors to resistance.

In the absence of a rigorous definition of tolerance for bacteriostatic antibiotics we hope that our study may allow others to think about this concept for drugs like trimethoprim. For greater clarity we have included this definition in the revised manuscript (Page 5, Line 160-165).

2. The authors only sequenced half of their evolved isolates (Figure 1), however in Figure 2D they show DHFR protein abundance for all 10 selected lines. They suggest that loss of mgrB confers trimethoprim tolerance by enhancing DHFR protein levels, however mgrB status (wild type versus mutant) should be established for all 10 lines. This would be simple to do with PCR and Sanger sequencing if needed (PCR alone would be sufficient to identify IS element insertions upstream of mgrB). Showing phenotype (i.e. protein abundance) data for all 10 lines but only genotype data for half of them, when it is very easy to genotype the other half, feels incomplete.

We agree that these data were incomplete in the manuscript. We have now amplified and Sanger sequenced the mgrB gene and its promoter from all 10 TMPR isolates. As expected, we do indeed find mutations at the mgrB promoter in all 10 isolates. These data have been added to the revised manuscript in the Figure 1A schematic.

3. Similarly, the authors should confirm rpoS mutations in lineages B/C in Figure 4 and Figure 6. Doing so would be easy with PCR and Sanger sequencing, and would further support their conclusions.

We thank the Reviewer for this suggestion. We have now PCR amplified rpoS from 3 resistant isolates each from the B and C lineages at 350 generations of evolution. We find that 5 of these have mutations at the rpoS locus. These data have been added to the revised version of the manuscript (Figure 4E, Page 10, Lines 317-320).

4. In panel 3B, the ΔphoP strain in a wild type background should be included as well, to remind readers that phoP deletion alone does not affect TMP MIC.

We have added these data to the revised manuscript.

5. In Figure 4, the trajectory of lineage A looks different from the other two lineages. The authors should make a note of this, and provide a justification for focusing on genetically characterizing this lineage rather than the other two.

We thank Reviewer 3 for this suggestion. We have included our rationale in the revised manuscript (Page 9, Line 303-305).

6. In Line 397, the authors should briefly explain the folA-independent mechanism of TMP tolerance in the lon-deficient *E. coli* strain.

We have shown earlier that Lon-deficient *E. coli* tolerate trimethoprim better due to overexpression of the acrAB efflux pump in this strain (Matange N., J. Bacteriol. 2020). We have mentioned this in the revised manuscript (Page 14, Line 418).